# Tissue-specific transcriptional profiling of plasmacytoid dendritic cells reveals a hyperactivated state in chronic SIV infection

**Michelle Y.-H. Lee**[1⦾], **Amit A. Upadhyay**[1⦾], **Hasse Walum**[1⦾], **Chi N. Chan**[2], **Reem A. Dawoud**[1⦾], **Christine Grech**[1⦾], **Justin L. Harper**[1], **Kirti A. Karunakaran**[1], **Sydney A. Nelson**[1], **Ernestine A. Mahar**[1], **Kyndal L. Goss**[1], **Diane G. Carnathan**[1], **Barbara Cervasi**[3], **Kiran Gill**[3], **Gregory K. Tharp**[4], **Elizabeth R. Wonderlich**[5], **Vijayakumar Velu**[1,6], **Simon M. Barratt-Boyes**[7], **Mirko Paiardini**[1,6], **Guido Silvestri**[1,6], **Jacob D. Estes**[2,8], **Steven E. Bosinger**[1,4,6]*

**1** Division of Microbiology & Immunology, Yerkes National Primate Research Center, Atlanta, Georgia, United States of America, **2** Vaccine and Gene Therapy Institute, Oregon Health & Science University, Beaverton, Oregon, United States of America, **3** Flow Cytometry Core, Emory Vaccine Center, Emory University, Atlanta, Georgia, United States of America, **4** Yerkes NHP Genomics Core Laboratory, Yerkes National Primate Research Center, Atlanta, Georgia, United States of America, **5** Southern Research, Frederick, Maryland, United States of America, **6** Department of Pathology and Laboratory Medicine, School of Medicine, Emory University, Atlanta, Georgia, United States of America, **7** Department of Infectious Diseases and Microbiology, Graduate School of Public Health, University of Pittsburgh, Pittsburgh, Pennsylvania, United States of America, **8** Oregon National Primate Research Center, Oregon Health & Science University, Beaverton, Oregon, United States of America

⦾ These authors contributed equally to this work.
* sbosing@emory.edu

**Data Availability Statement:** RNA-Seq data has been deposited in the Gene Expression Omnibus repository (National Center for Biotechnology

## Abstract

HIV associated immune activation (IA) is associated with increased morbidity in people living with HIV (PLWH) on antiretroviral therapy, and remains a barrier for strategies aimed at reducing the HIV reservoir. The underlying mechanisms of IA have not been definitively elucidated, however, persistent production of Type I IFNs and expression of ISGs is considered to be one of the primary factors. Plasmacytoid DCs (pDCs) are a major producer of Type I IFN during viral infections, and are highly immunomodulatory in acute HIV and SIV infection, however their role in chronic HIV/SIV infection has not been firmly established. Here, we performed a detailed transcriptomic characterization of pDCs in chronic SIV infection in rhesus macaques, and in sooty mangabeys, a natural host non-human primate (NHP) species that undergoes non-pathogenic SIV infection. We also investigated the immunostimulatory capacity of lymph node homing pDCs in chronic SIV infection by contrasting gene expression of pDCs isolated from lymph nodes with those from blood. We observed that pDCs in LNs, but not blood, produced high levels of IFNα transcripts, and upregulated gene expression programs consistent with T cell activation and exhaustion. We apply a novel strategy to catalogue uncharacterized surface molecules on pDCs, and identified the lymphoid exhaustion markers TIGIT and LAIR1 as highly expressed in SIV infection. pDCs from SIV-infected sooty mangabeys lacked the activation profile of ISG signatures observed in infected macaques. These data demonstrate that pDCs are a primary producer of Type I IFN in chronic SIV infection. Further, this study demonstrated that pDCs trafficking to LNs persist

Information database) under the accession
GSE178225.

**Funding:** This study was funded primarily by grant
R01 AI136990 from the National Institute of Health,
National Institute of Allergy and Infectious Diseases
(NIAID) awarded to S.E.B., and also supported by
R24 OD010445 (ORIP) from the National Institute
of Health, Office of The Director, National Institutes
of Health awarded to G.S., P51 OD 011132 from
the National Institute of Health, Office of The
Director awarded to the Yerkes National Primate
Center (Jonathan Lewin), S10 OD026799 from the
National Institute of Health, Office of The Director
awarded to S.E.B and the Yerkes Genomics Core,
grant R01 AI116379 from the NIAID awarded to M.
P., and the P30 AI050409 awarded to Carlos Del
Rio and the Center for AIDS Research (CFAR) at
Emory University. The work for this publication
was supported by NIH awards R01AI143411 (J.D.
E.) and P51OD011092 to Oregon National Primate
Research Center (Peter Barr-Gilliespie). The
funders had no role in study design, data collection
and analysis, decision to publish, or preparation of
the manuscript.

**Competing interests:** The authors have declared
that no competing interests exist.

in a highly activated state well into chronic infection. Collectively, these data identify pDCs as a highly immunomodulatory cell population in chronic SIV infection, and a putative therapeutic target to reduce immune activation.

## Author summary

For people living with HIV (PLWH), persistent immune activation is an obstacle to optimal health. In this study, we investigate the immunostimulatory potential of plasmacytoid dendritic cells in chronic SIV infection using comparative RNA-Seq. We observed that pDCs from SIV-infected rhesus macaques have highly activated profiles relative to uninfected animals; in contrast, pDCs from SIV-infected natural host sooty mangabeys had expression profiles similar to cells from uninfected animals. In chronically infected RMs, pDCs from lymph nodes maintained activation profiles elevated at levels even higher than those in the blood. Further, transcripts for the immunostimulatory cytokine family IFNA were readily detected in LN homing pDCs, but not those from blood. These data confirm pDCs as a major producer of Type I IFN in chronic SIV infection, and identify them as a target for immunotherapy.

## Introduction

At the end of 2019, approximately 38 million people were infected with HIV globally. Of these, an estimated 25.4 million had access to antiretroviral therapy (ART), however, 690,000 people still died of AIDS-related illnesses[1]. Despite these sobering statistics, efforts to distribute antiretroviral therapy and HIV prevention have made significant impact in the epidemic, reducing annual AIDS-related deaths by 60% compared to the peak of the epidemic. In light of these successes, efforts to expand the availability of ART are ongoing, and the proportion of people living with HIV that have access to treatment will continue to increase. Despite the tremendous success of modern ART at increasing the survival of infected individuals, life-long treatment of HIV infection raises additional challenges: (i) first, effective ART treatment of HIV infection requires life-long adherence, considerable monitoring, and remains expensive, which has hampered efforts to make access to ART broadly available in resource limited settings[1]; (ii) second, ART-treated individuals experience significantly higher morbidity compared to the general population, and these sequelae strongly correlate with inflammation[2]. For these reasons, the development of therapies that (i) enable HIV eradication or functional control of virus, or (ii) reduce the residual inflammation that remains after successful ART remain a priority for contemporary AIDS research.

In ART-treated HIV-infection[3] and in ART-naïve people living with HIV (PLWH)[4–7], disease progression is closely linked with inflammation and immune activation. Emerging evidence has also shown that persistent inflammation contributes to the persistence of HIV during suppressive ART[8,9]. The precise molecular causes of HIV-associated immune activation have not been definitively elucidated, however substantial experimental data has implicated the Type I interferon (IFN-I) system: in ART naïve patients IFNα predicts higher activation of CD8+ T cells[10], lower CD4+ T cell counts[10], and increased T cell apoptosis[11]. In the context of ART, elevated Interferon Stimulated Genes (ISGs) correlates with poor CD4+ T cell reconstitution[12]. A handful of studies have manipulated IFN-I in HIV-infected patients directly: administration of IFNα to ART-naïve patients[13] or during treatment interruption

[14] causes a decline in plasma viremia, but increases CD4+/CD8+ T cell activation[15] and reduces CD4+ T cell recovery[16]. Vaccination against endogenous IFNα in patients with end-stage AIDS led to stabilized CD4+ counts[17,18].

Several studies using the simian-immunodeficiency virus (SIV)/nonhuman primate (NHP) model have also shaped contemporary thinking about the role of Type I IFN in HIV infection. Our group and others have shown that during SIV infection, ISGs are rapidly shut off in non-pathogenic natural host NHP species, but persist indefinitely in pathogenically infected rhesus macaques (RMs)[19–23]. We also observed attenuated IFN responses in human viremic non-progressors, a rare phenotype of HIV-infection in which CD4+ T cell loss and overt disease is absent despite chronic high viremia[24]. In a series of studies manipulating the IFN-I system in SIV-infected macaques, we and others have shown that blockade of Type I IFN signaling in ART-naïve, acute infection accelerates mortality[25,26] but also reduced activation levels of CD4+ [25,26] and CD8+ T cells[26]. Conversely, IFN-I blockade in macaques with chronic, ART-treated SIV infection did not impact mortality and was generally well tolerated, however the impact on inflammation was restricted to mild reduction of ISGs, and no demonstrable impact on T cell activation[27]. Lastly, pilot experiments administering IFNα to ART-treated, SIV infected macaques have thus far demonstrated a minor, but unimpressive impact on reducing latent SIV associated with CD4+ T cells[28]. Contemporary models of the role of IFN-I in untreated HIV infection are in general agreement: IFN-I is beneficial to the host during acute SIV infection, by reducing viral burden and early dissemination and improving survival by suppressing virus during the generation of an adaptive immune response. However, in failing to clear virus, persistent IFN-I signaling during chronic SIV infection is hypothesized to drive pathogenic inflammation and immune activation. In the setting of ART treated HIV/SIV infection, IFN-I "intensification" efforts via exogenous administration have had little impact on the persistent reservoir in humans[14] and macaques[28]. Conversely, blockade of IFN-I have been able to reduce the levels of inducible and cell-associated HIV in ART-treated humanized mice[8,9], but data in NHP are inconclusive[27].

The IFN-I system remains one of the most tractable therapeutic targets for ameliorating harmful inflammation in the context of ART, and for reducing HIV that persists during ART treatment[29–32]. Despite its promise, definitive experimental and clinical data demonstrating the efficacy of antagonizing IFN-I for either inflammatory or cure endpoints are currently lacking, primarily for multiple reasons: (i) the primate type I IFN system is composed of 13 IFNA genes in humans[33], and at least 13 different IFNα subtypes in rhesus[34] that induce of hundreds of effector ISGs across diverse cell types[35]. Thus, the considerable redundancy and complexity of the IFN system represent a significant technical challenge when developing antagonistic therapies. (ii) Several important gaps remain in our understanding of IFN-I biology in persistent viral infections; specifically, whether the IFN-system is beneficial, dispensable or harmful in chronic viral infections[36], (iii) Lastly, identification of the primary cellular source of IFN-I during HIV/SIV infection remains elusive[32].

To address these gaps in knowledge, the goal of this study was to investigate the biology of plasmacytoid dendritic cells (pDCs) in the context of in vivo SIV infection using high-throughput transcriptomics. Plasmacytoid DCs are a unique subset of DCs, present at low frequencies (~0.1–0.5% of PBMCs) but producing >99% of host IFNα during viral infections [37]. Due to their central role in antiviral responses, pDC biology has been extensively studied in the context of HIV infection (reviewed in [38]). pDCs are depleted from the blood in HIV-infected patients[39–42] and do not recover to baseline after ART[43–49]. Concomitantly, defects in IFNα production have been observed in primary[50] and chronic HIV infection [51,52]. During acute SIV infection of macaques, absolute pDC numbers in the blood briefly increase, but these numbers quickly wane and steady-state levels are reduced relative to

uninfected individuals and baseline measurements[53,54]. The loss of pDCs from peripheral blood (PB) is likely due to redistribution and cell death, as during acute SIV infection, pDCs extensively relocate to lymph nodes where they upregulate apoptotic markers[55]. It is not clear whether the net effect of pDCs in HIV/SIV infection is protective (by helping to suppress virus) or harmful (by driving immune activation). While the loss of pDCs was originally considered to be a contributing factor to the immune dysfunction observed in AIDS[56], more recent data indicate that residual pDCs in HIV-infection are hyperfunctional[57] and promote CD8+ T cell activation[58]. In HIV and SIV infection, pDCs accumulate in the rectal[59–61] and vaginal[62] mucosa, where they recruit CD4+ T cells, stimulate ISG production and produce inflammatory cytokines.

Here, we addressed several questions pertaining to pDC biology in HIV infection: (i) First, to assess the contribution of pDCs to IFN-I production in chronic SIV infection. Although lymph-node homed pDCs clearly produce IFNα and IFNβ in acute SIV infection in macaques and natural host species [63], more recent studies have failed to observe IFNα in tissue pDCs during chronic infection[64]. (ii) Second, prior work by us and others have established that natural primate hosts of SIV infection resolve IFN-I responses after acute infection despite persistent, high-level viremia. To investigate the mechanism by which natural hosts avoid IFN-I signaling despite the prevalence of innate stimuli (i.e. continued SIV production), we conducted comparative transcriptomic analyses of pDCs from chronically infected rhesus macaques and sooty mangabeys. (iii) Lastly, as pDCs are increasingly becoming a therapeutic target for various immune-related diseases (transplant, autoimmunity, interferonopathies), we sought to create a compendium of the NHP pDC surface markers. Our data demonstrate that (i) during chronic SIV infection, lymph node derived pDCs produce high levels of IFNα transcripts; (ii) pDCs in lymph nodes of SIV+ RMs maintain a hyperactivated transcriptional profile relative to blood (iii) pDCs from the natural host species sooty mangabey maintain low levels of activation (iv) pDCs in chronically infected RMs express high surface levels of the exhaustion molecules TIGIT and LAIR1. This study identifies lymph node derived pDCs in SIV-infected RMs as hyperactivated, high-level producers of Type I IFN capable of driving immune activation during chronic SIV infection.

## Materials & methods

### Ethics statement

All animal experiments described were conducted following guidelines established by the Animal Welfare Act and by the NIH's Guide for the Care and Use of Laboratory Animals, 8th edition. All described procedures were performed following institutional regulations after review and approval by Emory University's Institutional Animal Care and Usage Committee (IACUC; Rhesus Macaques, Permit numbers: 201700044, 201700393, YER3000401, YER2003023, YER2002084, YER-2003576-ELMNTS, YER2002540, YER2002173, 201800100 and 201800243; Sooty Mangabeys, Permit number: 2002775) at Yerkes National Primate Research Center (YNPRC) or the University of Pittsburgh (PHS Assurance Number A3187-01 and IACUC protocol number 12040399). The SM study cohort at the YNPRC Main Center consisted of four SIV negative animals and four SIVsmm+ SMs. The four SIVsmm positive SMs acquired SIVsmm through experimental intravenous infection prior to November 2006 on previously approved IACUC protocols. Animal care facilities are accredited by the U.S. Department of Agriculture (USDA) and the Association for Assessment and Accreditation of Laboratory Animal Care (AAALAC) International. Approved procedures were put in place to ensure that potential distress, pain, discomfort and/or injury were limited to only those unavoidable in the conduct of the research plan. The sedative Ketamine (10 mg/kg) and/or

Telazol (4 mg/kg) were administered as necessary for blood and tissue collections with analgesics used when determined appropriate by veterinary medical staff. Euthanasia of RMs, performed at the end of the study by veterinary medical staff, was done using pentobarbital (100 mg/kg) under anesthesia and according to IACUC endpoint guidelines. RMs were fed standard monkey chow (Jumbo Monkey Diet 5037, Purina Mills, St Louis, MO) twice daily, and half an orange per day. Consumption is monitored and adjustments are made as necessary depending on sex, age, and weight so that animals receive sufficient food with minimum waste. SIV-infected RMs are singly caged but have visual, auditory, and olfactory contact with at least one social partner, permitting the expression of non-contact social behavior. The YNPRC enrichment plan employs several categories of enrichment five times per week where animals have access to more than one category of enrichment. IACUC proposals include a written scientific justification for any exclusions from some or all parts of the plan. Research-related exemptions are reviewed no less than annually. Clinically justified exemptions are reviewed more frequently by the attending veterinarian.

## Animals

The characteristics of the samples obtained from the non-human primates used in the transcriptomics and immunofluorescence microscropy (IF) components of this study are summarized in **S1 Table**. For RNA-Seq and IF experiments, samples were obtained from 16 Indian origin (IO) rhesus macaques (RMs) (*Macaca mulatta*), and eight sooty mangabeys (SMs) (*Cercocebus atys*) housed at the Yerkes National Primate Research Center (YNPRC). Three IO RMs housed at the University of Pittsburgh were used in the H7N3 flu virus exposure experiments. The twelve SIVmac239 infected RMs were challenged intrarectally (IR) at 10,000 TCID50 at the YNPRC. For the IF experiments, data are shown from two SIV+ YNPRC animals, and one SIV+ RM from the Oregon National Primate Research Center (ONPRC). Seventeen SIV-naïve male IO RMs were used for flow cytometry (TIGIT, CD4 and CLEC4C/BDCA-2) where all were Mamu-B*08/B*17 negative and six were Mamu-A*01 positive. Six male IO RMs were used for flow cytometry (TIGIT and PD-1) where all were Mamu-B*08/B*17 negative and two were Mamu-A*01 positive. These RMs were infected at 13–16 months with SIVmac239 IV at 300 TCID50. Eight RMs were used for flow cytometry (LAIR1) where all were Mamu-B*08/B*17 negative and 4 were Mamu-A*01 positive. These animals were infected IR with SHIV1175 at 700 TCID50 [65].

## Flow cytometry and cell sorting

Flow cytometry and sorting was performed using a flow cytometry panel described previously [66]. After Ficoll-based preparation, PBMCs were incubated with Live/Dead Fixable Aqua stain (Thermo Fisher Scientific) and then incubated with fluorescently labeled antibodies for cell surface staining. Lymph node mononuclear cells (LNMCs) were purified by mechanical disruption of lymph nodes and a single cell suspension was obtained by passing through a 70 μm cell strainer. For pDC and mDC staining, mononuclear cells were stained with the following monoclonal antibodies: anti-CD3 Pacific Blue (clone SP34-2), anti-CD16 BV650 (clone 3G8), anti HLA-DR PerCP-Cy5.5 (clone G46-6) and anti-CD11c APC (clone S-HCL-3) all from BD Biosciences; anti-CD20 Pacific Blue (clone 2H7), anti-CD4 BV650 (clone OKT4) all from BioLegend; anti-CD123 PE-Cy7 (clone 6H6) from Thermo Fisher Scientific, anti-CD14-ECD (clone RMO52) from Beckman Coulter and anti-BDCA-2 (clone AC144) from Miltenyi Biotec. pDCs were classified through the following gates for sorting: lymphocytes and monocytes, singlet, live cells, CD3-CD20-, HLA-DR+, CD14-, and CD11c-CD123+ and were sorted using a BD FACSAria II instrument (BD Biosciences) at the Emory Vaccine Center

Flow Cytometry Core at the YNPRC. mDCs were sorted as CD3-CD20-, HLA-DR+, CD14-, and CD11c+CD123- cells. pDCs and mDCs were sorted into 2 ml RNAse-free dolphin-nosed Eppendorf tubes containing RPMI, centrifuged at 600 x g for 10 min, media was aspirated to leave 50–100 μl of residual volume, and lysed by addition of 350 μl of RLT buffer (Qiagen) and stored at -80˚C until RNA extraction. For TIGIT and PD-1 staining, 18 parameter flow cytometry was performed on fresh peripheral blood mononuclear cells (PBMCs) using anti-human monoclonal antibodies (mAbs) clones that have been shown as being cross-reactive in rhesus macaques [67–69] in agreement with databases maintained by the NHP Reagent Resource (Mass Biologics): anti-CD21-PE-Cy7 (clone B-ly4), anti-Ki-67-AL700 (clone B56), anti-CD200-BV421 (clone MRC OX-104), anti-CD56-BV711 (clone B159), anti-HLA-DR-BV605 (clone G46-6), anti-CD16-BV650 (clone 3G8), anti-CD3-BUV395 (clone SP34-2), anti-CD8-BUV496 (clone RPA-T8), and anti-CD95-BUV737 (clone DX2) all from BD Biosciences; anti-CD123-FITC (clone 6H6), anti-CD4-APC-Cy7 (clone OKT4), anti-CD20-PerCP-Cy5.5 (clone 2H7), and anti-PD-1-BV785 (clone EH12.2H7) all from Biolegend; anti-TIGIT-PE (clone MBSA43) from eBioscience; anti-CD14-ECD (clone RMO52), anti-CD27-PE-Cy5 (clone 1A4CD27), anti-NKG2a-APC (clone Z199) all from Beckman Coulter; and Live/Dead Fixable Aqua viability dye from Thermo Fisher. Cells were fixed and permeabilized with BD Cytofix/Cytoperm. Acquisition was performed on a minimum of 120,000 stopping events (i.e. Live CD3+ T-cells) on a LSR Fortessa (BD Biosciences) using FACS DiVa software. For staining of LAIR1, whole blood (100 μl) were surface stained for 30 minutes followed by treatment with 1x BD FACS lysing solution (BD Biosciences) for 10 minutes at room temperature. After a final wash, cells were assessed by flow cytometry as described previously [65]. Antibody clones included anti-CD3 (clone SP-34-2), anti-CD4 (clone OKT4), anti-CD8 (clone SK1), anti-CD20 (clone 2H7), anti-CD11C (clone S-HCL-3), anti-CD14 (clone M5E2), anti-CD123 (clone 7G3) and anti-LAIR1 (clone DX26). All flow cytometry data were analyzed using FlowJo software (TreeStar).

## Plasma SIV viral load assay

Plasma SIV viral loads were conducted in the Translational Virology Core of the Center for AIDS Research at Emory University using a standard quantitative PCR (qPCR) assay with a limit of detection of 60 copies per milliliter of plasma [28].

## Ex vivo TLR7 stimulations and ELISPOT

ELISpot assays were conducted using the Human IFN-α2 ELISpot BASIC (ALP) kit (Mabtech) following manufacturer's instructions. Before seeding of cells, plates were pre-coated with coating antibody. Sorted pDCs were added to wells containing $2.5x10^5$ non-sorted PBMCs at 10–500 pDCs per well. 1000 non-pDCs (lymphocyte, singlet, live cells, CD3-CD20-, HLA-DR+, CD14-, CD123-) were added into a separate set of wells. Cells were incubated with 2.5 μg/mL of the TLR7/8 agonist, CL097 (InvivoGen) and incubated for 48 hours at 37˚C. The plate was incubated with the biotinylated detection antibody for 2 hours at room temperature followed by incubation with Streptavidin-ALP for 1 hour at room temperature. BCIP/NBT (Mabtech) was then added to the plate, which was allowed to develop for 15 minutes. Plates were scanned using the Immunospot Analyzer (Cellular Technology Limited) and spots counted with ImageJ.

## Ex vivo H7N3 influenza virus infection of purified pDCs

pDCs were sorted from cryopreserved spleen cell suspensions from three SIV-naïve rhesus macaques. pDCs were cultured (cell counts ranged from 27,550 to 83,000) for 7 hours in

complete RPMI-1640 alone or with H7N3 influenza virus at a multiplicity of infection of five. After stimulation, cells were pelleted by low speed centrifugation, the media was aspirated and pellets were lysed in 350 μl of RLT buffer supplemented with 1% BME.

## Multiplex immunofluorescence

Heat induced epitope retrieval was performed on 5 μm formaldehyde fixed, paraffin-embedded tissue sections from lymph nodes of SIV-chronically infected monkeys using Tris pH8.6 buffer in a Biocare Decloaker (110˚C for 30 minutes). This is followed by blocking with 0.25% Casein for 20 minutes at room temperature and overnight incubation with the mouse anti-IFN-α antibody clone MMHA-2 (PBL Assay Science). Endogenous peroxidases were then inactivated by incubation in 1.5% $H_2O_2$ for 5 minutes at room temperature. The slides were then incubated with the Polink-1 HRP polymer detection system for mouse antibody (GBI Labs) for 1 hour at room temperature, followed by signal development using Tyramide CF-568 (Biotium). Stripping of bound antibodies was then performed by boiling in Tris pH8.6 retrieval buffer for 25 minutes. The slides were then incubated with a Goat anti-CD303 polyclonal antibody (R&D Systems) overnight at room temperature, followed by detection with an DyLight-755 conjugated donkey anti-Goat antibody (Invitrogen). Tissues were counterstained with DAPI (Invitrogen, 500 ng/mL) and cover slipped with #1.5 GOLD SEAL cover glass (EMS) using Prolong Gold reagent (ThermoFisher). Whole-slide high-resolution fluorescent scans were performed using a Plan-Apochromat 20X objective (NA 0.80) using the Zeiss AxioScan Z.1 slide scanner. DAPI, AF488, AF568, and Cy7 (For DyLight-755) channels were used to acquire images. The exposure time for image acquisition was between 4 and 300 ms.

## RNA sequencing

RNA Extraction and NGS library preparation was performed in the Yerkes NHP Genomics Core (http://www.yerkes.emory.edu/nhp_genomics_core/). RNA extraction and library preparation was performed using previously published methodology. Briefly, RNA was isolated using RNeasy Micro Kits (Qiagen) with on-column DNase Digestion (Qiagen). RNA quality and quantity were determined using a Bioanalyzer RNA Pico chip (Agilent). RNA was converted to cDNA and amplified using the Clontech SMART-Seq v4 Ultra Low Input RNA kit (Takara Bio) according to the manufacturer's instructions. Amplified cDNA was fragmented and appended with dual-indexed bar codes using the NexteraXT DNA Library Preparation kit (Illumina). Libraries were validated by capillary electrophoresis on an Agilent 4200 TapeStation, pooled at equimolar concentrations, and sequenced on an Illumina HiSeq3000 at 100SR, yielding 20–25 million reads per sample.

Alignment was performed using STAR version 2.5.2b [70]. Transcripts were annotated using the MacaM assembly and annotation of the Indian rhesus macaque genome [71], (http://www.unmc.edu/rhesusgenechip/index.htm#NewRhesusGenome).

## Bioinformatics and data analysis

The quality of sequenced reads was assessed with FastQC (https://www.bioinformatics. babraham.ac.uk/projects/fastqc/). The Spliced Transcript Alignment to a Reference (STAR) [70] aligner version 2.5 was used to align raw reads to build version 7.8.2 of the MacaM genome reference [71] (http://www.unmc.edu/rhesusgenechip/index.htm#NewRhesusGenome), kindly provided by Dr. R Norgren, University of Nebraska. Transcript abundance estimates were calculated internal to the STAR aligner using the algorithm of htseq-count [72]. DESeq2 was used for normalization [73], producing both a normalized read count table and a regularized log expression table, and to calculate counts for basic annotated genes. For figures showing the

surface protein and transcription factor genes ranked by expression, reads were aligned to both the MacaM reference and build version 38 of the human genome (GRCh38). In this case, raw reads were aligned to a human reference, even though the studied organisms were non-human primates, because the human genome provides better coverage and/or more of the immune related genes described. Parallel analyses of the surface protein and transcription factor gene-ranking plots using data aligned to the MacaM reference can be found in the **S3 and S6** Figs. Principal components analysis (PCA) was performed on log transformed counts, generated using the "regularized log" method from the DESeq2 package. Normalization and differential gene expression analysis were carried out using the DESeq2 package [73] in R 3.5.1. Genes were considered differentially expressed with an adjusted p-value (FDR) lower than 0.05. Samples used for comparing pDC gene expression in PBMCs and LNs were sequenced in two different batches, and batch correction was performed by including "batch" as a covariate in the DESeq2 models.

For obtaining the normalized counts for reads mapping to SIV genome, a new reference was built by combining the MacaM reference with ERCC (https://assets.thermofisher.com/TFS-Assets/LSG/manuals/ERCC92.zip) and SIVmac239 (NCBI Accession: M33262) sequences. The gff3 file for SIV was converted to gtf format using gffread utility (https://github.com/gpertea/gffread). The CDS entries in the resulting gtf were modified to exon and a concatenated gtf file was made for MacaM, ERCC and SIV. STAR was used to generate a composite reference to which the reads were aligned. All reads mapping to the SIV genome were summed and counts per million were determined for each sample.

Gene set enrichment analysis was performed using the GSEA software [74] version 3.0. DESeq2 regularized log transformed counts were used as input for the GSEA analyses. Batch correction was done using the removeBatchEffect method from the limma R package with log transformed values used as input. Gene sets representing the activity of pathways were derived from the Molecular signature database [75]. For GSEA analyses comparing SMs and RMs, gene-set permutation was used for statistical inference due to small sample size. Gene sets were considered statistically up- or down regulated across phenotype groups with an adjusted p-value (FDR) less than 0.05. Ingenuity Pathway Analysis (IPA) software was used to identify gene sets enriched for genes differentially expressed between PBMC and LN samples. We picked genes to include in the IPA analysis based on both statistical significance (FDR<0.05) and effect size (Fold Change > 1.5) from the DESeq2 analyses.

All figure items were created using the ggplot2 package in R. Heat maps were generated using regularized log values from DESeq2 and normalized using baseline (SIV-/PBMC) group median subtraction. All plots showing "Normalized read counts" use read values following "size factor normalization" from the DESeq2 package.

Graphs and statistical analysis for flow cytometry and ELISPOT were generated using PRISM (GraphPad Software).

## Results

### Experimental design

An overview of the experiment is shown in **Fig 1A**, and detailed metadata for all animals used in this study is contained in **S1 Table**. To profile the transcriptional response in pDCs in the context of natural SIV infection, we sorted pDCs from the peripheral blood (PB) of four SIV-negative SMs and four SMs chronically infected with SIVsmm (9–12 years infected, median plasma viral load (PVL) of $2.58 \times 10^4$ copies/mL). To directly contrast these data with transcriptional profiles from pDCs in pathogenic SIV infection, we also sorted PB pDCs from four uninfected, and four RM chronically infected with SIVmac239 (duration 179–237 days,

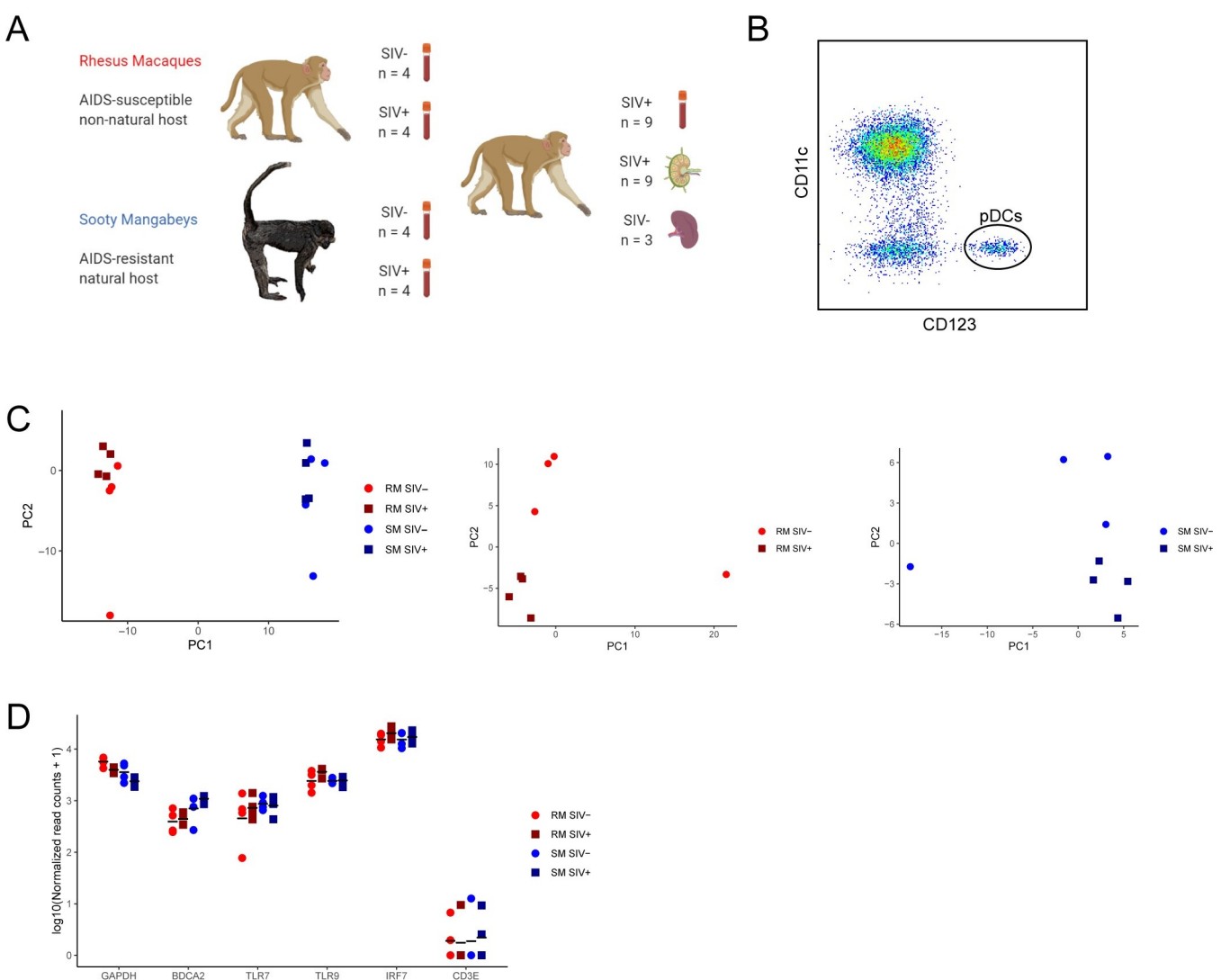

**Fig 1. Overview of comparative transcriptomics of pDCs in SIV infection. (A)** Summary of the samples used to compare the transcriptomes of pDCs sorted from peripheral blood in SIVsmm infected sooty mangabeys and SIVmac239 infected rhesus macaques or compare the transcriptomes of pDCs sorted from the peripheral blood and lymph nodes of SIVmac239 infected rhesus macaques. No animals were on ART. (created with BioRender.com). **(B)** Representative flow plot of sorting strategy to obtain pDCs from PBMCs, based on CD11c and CD123. **(C)** Principal component analyses (PCA) of transcriptome from pDCs, with SIV- RMs shown as red circles, SIV+ RMs as red squares, SIV- SMs as blue circles, and SIV+ SMs as blue squares. **(D)** Normalized read counts (log10) for four representative genes with known expression in pDCs and a housekeeping gene, GAPDH. The normalized read counts of the CD3E gene, which is primarily expressed primarily by T lymphocytes, is also included to show background read count levels. Dots show read counts for individual animals and black lines show mean normalized read counts (log10) of n = 4 for each experimental group.

median PVL of 2.39x10$^5$ copies/mL). A second set of samples, derived from ten SIV-infected RMs (including the four chronically infected RM above), was obtained at the time of sacrifice for comparisons of gene expression in pDCs in peripheral blood mononuclear cells (PBMCs) compared to pDCs from lymph nodes (LNs). Eight sets of PB and LN samples were matched, whereas two RMs each contributed only a PBMC or LN sample. These animals were sampled at 97–370 days post-infection with a median PVL of 3.16x10$^5$ copies/mL. Gene expression in pDCs in the PB and LN of all nine SIV-infected RMs was used to create cell surface marker and transcription factor compendiums. Finally, a third set of RNA samples were derived from pDCs isolated from the spleens of three SIV-naïve RMs that were exposed to H7N3 influenza

virus in vitro. For IF, axillary LNs were obtained at necropsy from three RMs chronically infected with SIVmac239. None of the SIV-infected animals were on ART at the time of sample collections. A representative scatter plot of the final gate based on CD11c and CD123 used in flow analysis of pDCs is shown in **Fig 1B,** with the full gating strategy shown in **S1 Fig**. Depending on the sample, yield of pDCs ranged from 1,728–28,793 (PBMC) and 3,543–37,732 (single or pooled LNs), with an average purity after sorting exceeding 99%. Principal component analysis of the transcriptional dataset for SIV-/SIV+ RM and SM showed large inter-species variation in gene expression, and clustering of samples by infection status in RMs (**Fig 1C**). To validate the sensitivity and accuracy of the RNA-Seq data to describe pDC biology, we examined the average gene expression of several canonical pDC genes relative to lineage markers from other lymphocyte subsets. As shown in **Fig 1D**, the expression of BDCA-2/CD303/ CLEC4C, TLR7, TLR9 and IRF7 were all detected with read counts at levels equal to, or in the range of the housekeeping gene GAPDH. In comparison, read counts for the TCR subunit CD3E, was only marginally above background levels. These data indicate that the RNA-Seq experiment was highly specific for pDC specific features.

## pDCs in SIV natural hosts do not have chronic ISG expression

In previous work, we and others have demonstrated that natural host species such as sooty mangabeys do not have a detectable Type I Interferon response during chronic SIV infection, although they have a robust ISG response during the acute phase [19]. Our prior studies examined gene expression profiles in whole blood and lymph node biopsies using microarray technology [19]. Similarly others have made concordant observations in the SIV natural host African Green Monkey (*Chlorocebus aethiops*), observing expression of ISGs during acute infection in peripheral blood and lymph node CD4+ T cells as well as total cells from colon and lymph node biopsies [20–22]. To date, it has not been determined if the lack of an ISG signature observed in natural hosts, or if the persistence of ISGs observed in pathogenic lentiviral infection, extends to other immune cell subsets beyond T cells. To assess ISG expression in pDC transcriptomes, we examined ISG expression in several ways: first, we measured expression of a 19-gene ISG signature panel representative of Type I IFN signaling we have described previously as being upregulated in RMs after SIVmac239 infection [25,27] and sensitive to inhibition by in vivo antagonism of the IFNAR [76]. Summing of the read counts for all 19 ISGs showed a striking upregulation of ISGs in PB pDCs from SIV-infected RMs relative to uninfected animals (p = 0.0286) (**Fig 2A**). In contrast, we did not observe a difference in the summed ISG read counts in pDCs derived from infected SMs relative to their uninfected counterparts (p = 0.8857). Examination of individual genes demonstrated that there was consistent upregulation of the majority of ISGs in all four SIV-infected RMs, indicating that the ISG signature was not due to a few highly expressed genes (**Fig 2B**). In the four sampled SIV-infected SMs, individual ISGs in pDCs did not show consistent upregulation. Since pDCs are the primary source of IFN-I in response to SIV, we determined the ISG response profile in non-pDCs (myeloid dendritic cells, mDCs, same gating as pDCs except CD123-CD11c+, **S1 Fig**) that were sorted from the same animals in parallel with pDCs. Consistent with the data from pDCs, a strong upregulation in all ISGs in our panel was seen in mDCs derived from SIV-infected RMs, but not SMs (**S2 Fig**).

A limitation of this analysis was that it was underpowered, at n = 4 for each experimental group. To test if the observed upregulation of ISGs was statistically significant, we used gene-set enrichment analysis (GSEA) as it tests for the cumulative effect of multiple genes. To ensure our analysis was not biased due to our selection of ISGs, we tested for upregulation of a highly used, standardized geneset from the "Hallmark" module of the Molecular Signatures Database

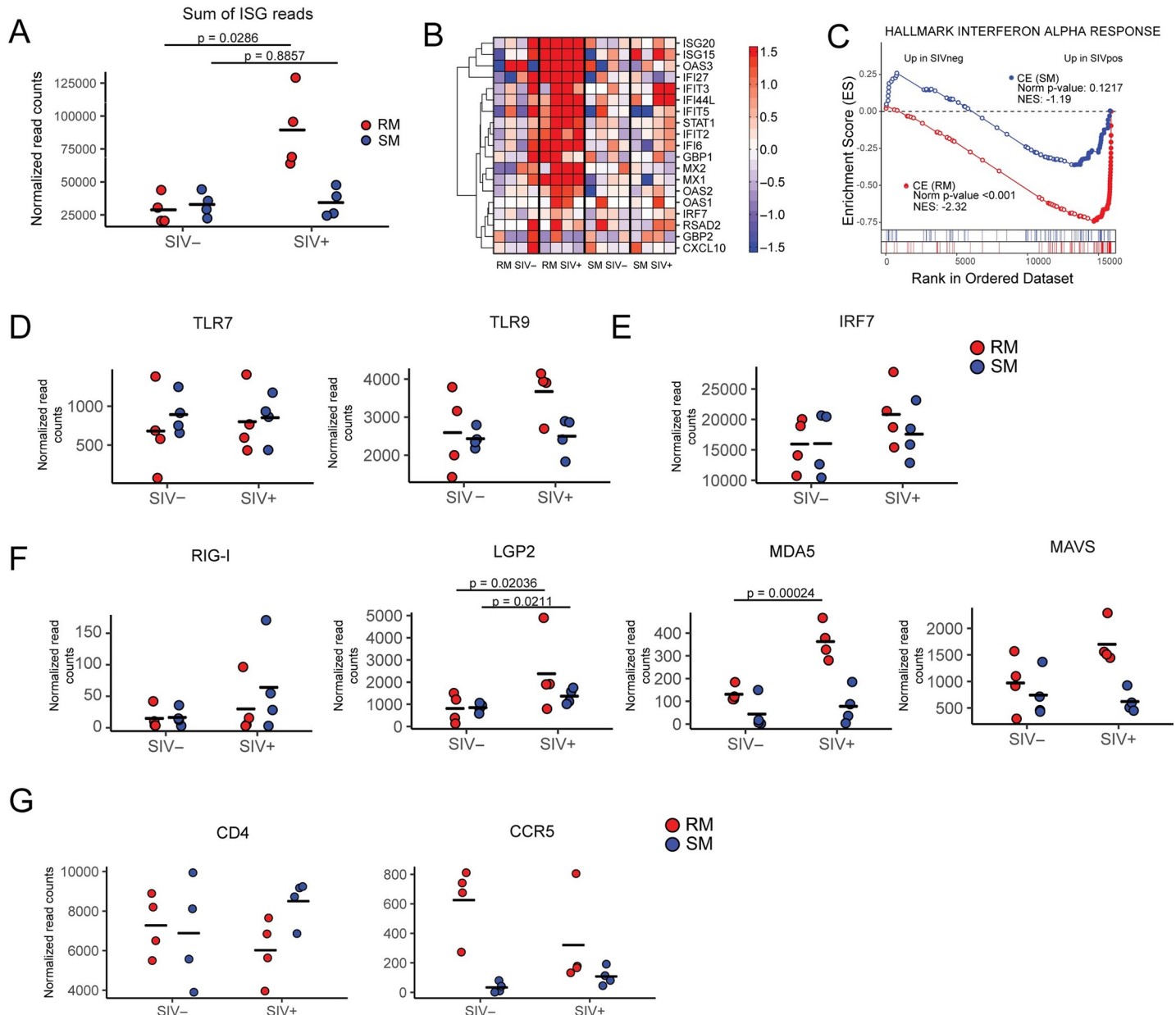

**Fig 2. Persistent expression of interferon stimulated genes (ISGs) in peripheral blood pDCs in chronic SIV infection of RMs but not SMs. (A)** Sum of all normalized read counts mapping to 19 ISGs known to be modulated in SIV infection of RMs [27]. **(B)** Heat map of the 19 ISGs used in panel A. The values shown were first transformed using the "rlog" method from DESeq2, data were normalized by subtraction of the median (on a per-gene basis) of the baseline samples (i.e. SIV- RM and SIV-SM) from each datapoint; RMs data was normalized by the SIV-RM median and SM data by the SIV-SM median. The color scale of the heatmap is set to maximal at fold-changes of -1.5 and 1.5. **(C)** GSEA enrichment plot of the Hallmark interferon alpha response, (MSIGDB geneset #M5911) for contrasting expression in SIV- vs SIV+ for RMs and SMs, in two separate analyses depicted by the red and blue lines, respectively. This plot panel also shows NES and nominal p-values. Normalised read counts for the endosomal pattern recognition receptors, TLR7 and TLR9 **(D)**, IFN signaling gene IRF7 **(E)** cytosolic pattern recognition receptors and associated signaling proteins, RIG-I, LGP2, MDA5 and MAVS **(F)** in uninfected and SIV-infected RM and SM peripheral blood pDCs. **(G)** Normalised read counts for the SIV receptor CD4 and co-receptor CCR5 contrasting SIV- and SIV+ RMs and SMs. All plots show values for individual RMs and SMs and black line indicates mean, n = 4 for each condition. "Normalized" refers to size factor normalization from DESeq2. In **(A)**, statistical significance was determined using a Mann Whitney test, comparing infected and non-infected samples from the same species. In **(D)**–**(G)**, only significant p-values from DESeq2 analysis is shown.

(HALLMARK INTERFERON ALPHA RESPONSE– https://www.gsea-msigdb.org/gsea/msigdb/). We observed statistically significant enrichment of this IFNA regulated geneset (p <0.001) when we contrasted transcriptomes from SIV-infected RMs relative to uninfected

RMs (**Fig 2C, red curve**). We did not observe significant enrichment of the HALLMARK IFNA geneset when contrasting transcriptomic data from SIV-infected versus uninfected SM samples (**Fig 2C, blue curve**). Collectively, these data extend previous observations of the ISG response in pathogenic, non-natural SIV infections to include pDCs as an immune subset subject to activation effects of Type I IFN. Moreover, the data provide additional evidence, in differing animals from our original study, and in a highly defined immune subset, that the natural host NHP species SM does not exhibit chronic IFN activation despite persistently high levels of plasma virus.

We next examined the expression of antiviral pathogen-recognition receptors (PRRs) in pDCs. The expression of TLR7 was moderate, at mean expression of 771.8 and 809.1 in SIV naive RM and SM, respectively. (**Fig 2D**). Levels of TLR7 were not significantly elevated after SIV infection in either species. TLR9 was more highly expressed than TLR7 in all conditions (at least 2.7-fold difference) and where there was no change in TLR7 in SIV infection, there was a 1.43-fold upregulation of TLR9 in SIV+ RMs (**Fig 2D**), although this did not reach significance. As expected, very high levels of IRF7 mRNA were observed in pDCs, with average read counts exceeding 15,000 in all groups, however a strong upregulation was only observed in samples from chronically infected RMs (**Fig 2E**). Read counts of all other TLRs were no higher than 400 in all samples (**S3 Fig**). While the activity of TLR7 and TLR9 to respond to viral RNA and DNA is well established for pDCs, it is unclear if RIG-I-like receptors (RLRs) are utilized in this lineage. To assess if the levels of RLRs could account for the differential ISG response, we examined the expression of RIG-I, LGP2, MDA5 and the adaptor molecule MAVS. Read counts of RIG-I, LGP2 and MDA5 were lower than that of TLR9, and although MDA5 is shown to be upregulated in response to SIV infection in RMs (**Fig 2F**); the levels are 10-fold lower than that of TLR9. Of note, the expression of RIG-I was particularly low, with average expression levels below 25 read counts for uninfected samples (**Fig 2F**). The downstream adaptor molecule of the cytosolic PRRs, MAVS is at levels nowhere near that of the downstream transcription factor, IRF7 even though it is upregulated in response to SIV infection in RMs. These observations suggest a minor role for or a lack of RIG-I associated IFN-I signaling in pDCs in response to SIV infection.

To test for potential molecular differences between SMs and RMs that facilitate the distinct outcomes in ISG production, we also considered expression levels of the primary receptor and co-receptor for SIV, CD4 and CCR5, respectively. CD4 was expressed at high levels on pDCs, with average read counts of 6278 to 8212 across all samples (**Fig 2G**). The levels of CD4 expression, however, did not differ markedly between species, regardless of infection status. This transcriptomic data is supported by flow cytometry, where CD4 is highly expressed on the surface of PB pDCs from 15 uninfected RMs, which is comparable to the expression of BDCA-2/CD303/CLEC4C, a pDC specific cell-surface marker (**S4A Fig**). We also compared the expression of the SIV co-receptor CCR5 between species, as previous reports using flow cytometry have described reduced frequency of CCR5+ pDCs in SMs and AGMs relative to RMs[77]. Consistent with these prior flow-cytometry based studies, we observed that mRNA levels of CCR5 were substantially higher in RMs than SMs in SIV- animals (**Fig 2G**). In SIV infection of RMs however, CCR5 was downregulated and in three out of four SIV+ RMs, the frequencies of CCR5-positive pDCs were indistinguishable from that of SMs. The lower expression of CCR5 in SIV-infected RMs may be due to the emigration of CCR5hi pDCs from the periphery into tissues. Collectively, these data demonstrate that pDCs from both RMs and SMs maintain high levels of CD4, but CCR5 expression is lower in uninfected SMs.

## pDCs in LNs of RMs express high levels of IFNα and IFNβ during chronic SIV infection

The unabated activation of the Type I IFN system has long been considered to be one of the major underlying causes of immune activation and disease in HIV infection [27,31]. While ISGs are easily detectable in SIV infection, the cellular source of Type I IFN (i.e. IFNα/β/ω) remains unknown. pDCs have long been hypothesized to be a major producer of Type I IFN in response to SIV infection, however, prior studies in macaques have suggested their contribution to IFN-I production is limited to the acute phase of infection [64]. Here, we performed a paired analysis of expression in sorted RM pDCs from peripheral blood (n = 9) and pooled lymph node (n = 9) for Type I IFN transcripts. To maximize detection of IFNA transcripts, which is represented by 9 IFNA genes in the MacaM reference for Indian RM, we summed the read counts for all genes. As shown in **Fig 3A**, the cumulative sum of all IFNA transcripts in pDCs sorted from PBMCs in SIV+ RMs was undetectable. In contrast, when we analyzed pDC RNA samples from the LNs of the same animals, we observed a mean cumulative sum read count of 563.9 for IFNA transcripts. Of the 9 animals assayed, we observed high levels of IFNA transcripts in the LN of 7 animals. To put our expression level of IFNA into a biological context, we isolated splenic pDCs from three SIV-naïve RMs and exposed them to H7N3 influenza virus in vitro. As shown in **Fig 3A**, we did not observe any read counts for IFNA in the unstimulated pDCs, whereas we observed summed IFNA read counts of 2887, 37026, and 93942 in stimulated samples. These data were important for two reasons: first, they demonstrated that freshly isolated lymphoid-tissue derived pDCs from uninfected animals do not have detectable IFNA transcripts due to tissue localization; second, these data demonstrate that while lower, IFNA transcripts detected in pDCs analyzed directly ex vivo from SIV chronically infected LNs were comparable to levels observed in pDCs after acute exposure to high titres of H7N3 influenza virus, even though the cell input was approximately 10-fold lower. When examining the read counts for the individual IFNA subtypes in SIV-infected RM, all showed a significantly higher level of expression in LN pDCs relative to matched PB samples, in which the majority of samples had undetectable levels of IFNA (**Fig 3C**). Similarly, when examining PB pDCs of SIV-infected RMs, IFNB transcripts are undetectable but were significantly elevated (p = 0.029) in LNs at levels comparable to the individual IFNA subtypes (**Fig 3B**). Exposure of splenic pDCs to H7N3 influenza virus in vitro also showed high upregulation of IFNB transcripts (p = $4.8 \times 10^{-6}$) (**Fig 3B**). To confirm the production of IFN-α in LN-derived pDCs at the protein level, we performed multiplex staining of LN samples from another cohort of three chronically infected RMs for CD303/BDCA-2 and IFN-α with fluorescently labelled antibodies. We observed that the majority of CD303+ cells were localized in the extra-follicular, medullary cord regions of the LN and that there was a strong co-localization of CD303+ and IFN-α cells (**Fig 3D**). These data demonstrate that in chronic untreated SIV infection, there is production of IFN-α protein from pDCs in the LN.

One potential technical concern with observation of LN-derived pDCs producing IFNA mRNA is that it is formally possible that the source of IFN is a non-pDC contaminant. To test this possibility, we assessed the contribution of IFN-α production in response to TLR7 stimulation in total PBMCs by spiking in pDCs or non-pDCs. pDCs were sorted from PB (gating strategy, **S1 Fig**) and added in increasing frequency from 10–500 into wells containing $2.5 \times 10^5$ PBMCs. The number of IFN-α producing cells detected after treatment with the TLR7/8 agonist CL097 was compared to wells containing no additional pDCs or wells containing 1000 non-pDCs (same gating strategy for pDCs, except CD123-, **S1 Fig**). There was a three-fold increase in IFN-α producing cells after addition of 500 pDCs whereas no change was observed when 1000 non-pDCs were added (**Fig 3E**). These data demonstrated that pDCs are the

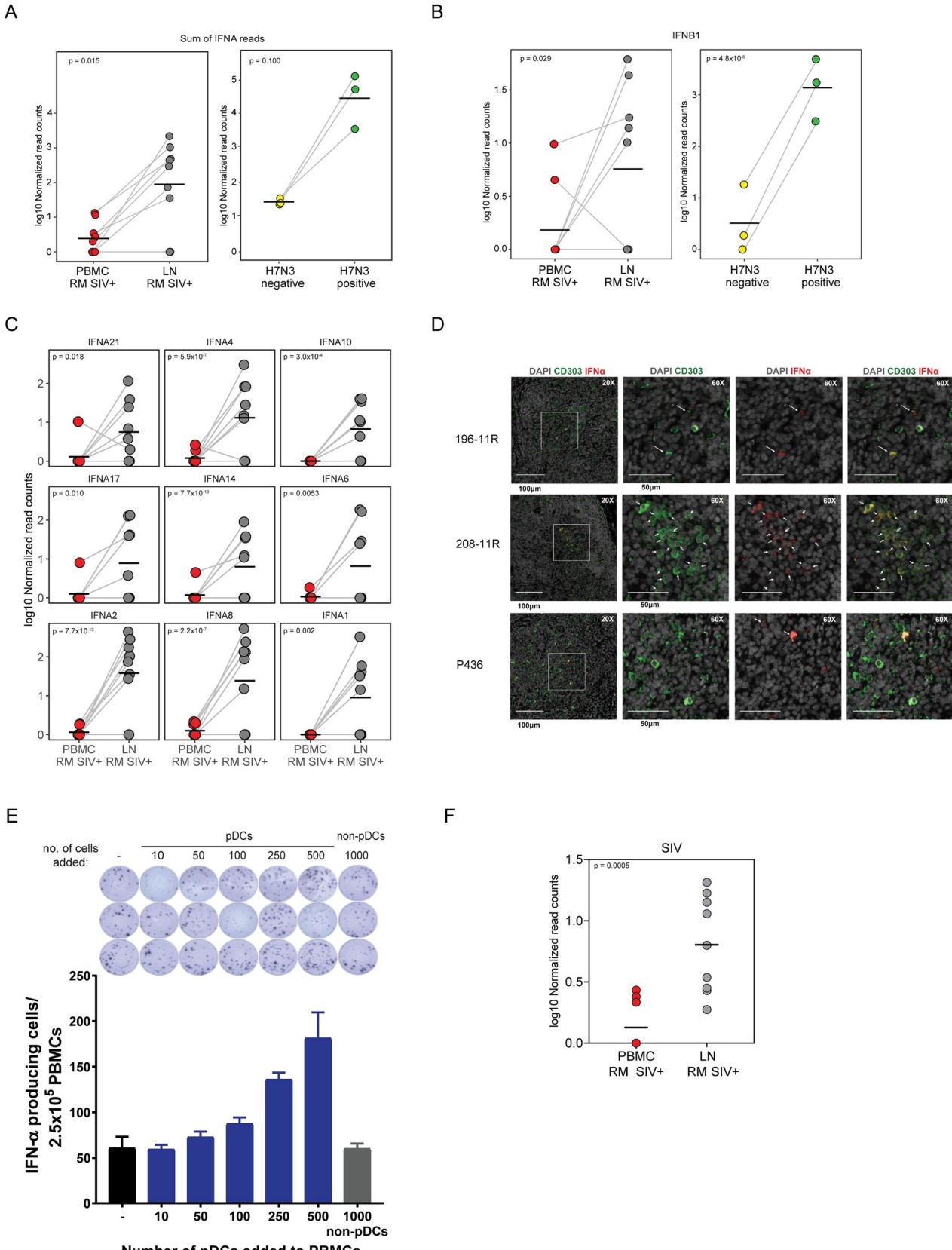

**Fig 3. High levels of IFNA are produced by pDCs in the lymph nodes but not the peripheral blood of SIV-infected macaques.** (**A**) Log10 normalized summed read counts of transcripts from all IFNA subtypes in pDCs isolated from the blood (red) or lymph node (grey) of matched and unmatched SIV+ RMs (left panel) or in untreated (yellow) or H7N3 influenza exposed (green) splenic pDCs isolated from SIV-naïve RMs (right panel). (**B**) Log10 normalized read counts of IFNB1 transcripts in pDCs isolated from the blood (red) or lymph node (grey) of matched and unmatched SIV+ RMs (left panel) or in untreated (yellow) or H7N3 influenza exposed (green) splenic pDCs isolated from SIV-naïve RMs (right panel). (**C**) Log10 normalized read counts of transcripts for 9 individual IFNA genes in pDCs isolated from the blood (red) or lymph node (grey) of matched and unmatched SIV+ RMs. (**D**) Multiplex immunofluorescence staining (20X, 60X) of IFN-α (red) and CD303 (green) in axillary lymph node sections from RMs chronically infected with SIV. Arrows identify cells that show co-localization of CD303 and IFN-α, indicative of pDCs. DAPI staining is shown in grey. (**E**) pDCs were sorted from RM PBMCs and added, at the numbers specified in the figure, to triplicate wells that contained $2.5 \times 10^5$ unsorted PBMCs. 1000 non-pDCs (CD3- CD20- HLA-DR+ CD14- CD123- lymphocytes/monocytes) were added to a seventh set of triplicate wells. Cells were cultured for 48 hours in the presence of the TLR7/8 agonist, CL097 at 2.5 μg/mL in plates pre-coated with anti-IFN-α antibody. Production of IFN-α was determined by ELISpot assay. Bars show mean +/- SD from the three wells of one independent experiment. (**F**) Log10 normalized read counts of SIV transcripts mapped to SIVmac239 in pDCs isolated from the blood (red) or lymph node (grey) of matched and unmatched SIV+ RMs. In panels **A-C** and **F**, read counts are shown for individual RMs and the black bar indicates mean. Statistical significance in (**A**) and (**F**) were determined using a Mann Whitney test. P-values in (**B**) and (**C**) are derived from DESeq2 analysis.

primary producer of IFN-α, and that the contribution of non-pDCs to IFN-α production is negligible.

Lastly, we investigated if the level of cell-associated SIV RNA was different between LN and PB pDCs by quantitating the normalized RNA-Seq reads mapping to a genomic reference of SIVmac239. As shown in **Fig 3F**, we observed that samples from LN harboured a significantly higher number of reads mapping to SIV (median log10 normalized read counts LN: 0.799 vs PBMC: 0, p = 0.0005). This observation was consistent when quantifying raw, non-normalized SIV read counts (**Fig 4B**), and SIV read count was independent of the input number of pDCs (**Fig 4C**).

Collectively, these data demonstrate that while IFNA cannot be detected in pDCs in peripheral blood, pDCs in lymphoid tissue during chronic SIV infection produce high levels of IFNA.

## pDCs in lymph nodes during chronic SIV infection acquire an activated, immunostimulatory state

Previous work by others have used microarrays to characterize the transcriptome of pDCs in people living with HIV [57]. While these studies detailed cumulative differences observable by PCA, they did not detect significant DEGs between the pDCs from primary HIV-infected patients and uninfected donors [57]. In prior studies, we and others have observed that after SIV infection, pDCs accumulate in the LN, vagina and rectal mucosa of RMs [59,60,62] and humans [61]. These prior anatomical observations, combined with our observations here that LN-derived pDCs, but not those in peripheral blood, produce IFNA, suggest that tissue-derived pDCs have an elevated activation profile relative to those in blood. To examine this at the transcriptional level, we contrasted expression in pDCs isolated from PB and LN of SIV-infected RMs (n = 9). This analysis yielded 2771 differentially expressed genes (DEGs) (padj < 0.05) (**Fig 4**, complete DEG list in **S2 Table**). In the PCA analysis, there is an observable clustering of the transcriptome of pDCs isolated from LN distinct to those from PBMCs in SIV-infected RMs (**Fig 4A**). We observed that the read counts of CCR7 were significantly higher in the LN-derived pDCs compared to the PB pDCs, consistent with these cells homing to LNs (median read count, LN: 3181 vs PBMC: 562, 4.3-fold induction, padj = 0.0029) (**Fig 4B**). As shown in the heatmap of the top 500 DEGs (**Fig 4C**), when compared to a gene-centric mean, the majority of genes exhibited higher or lower expression in lymph nodes, indicating that they were more transcriptionally active than the sample from blood. To examine if interferon stimulated genes were significantly upregulated in LN pDCs, we tested for enrichment using a 19-gene panel of ISGs we have consistently observed as upregulated in prior studies using rhesus macaques [25,27,28]. As shown in **Fig 4D**, although blood-derived pDCs had

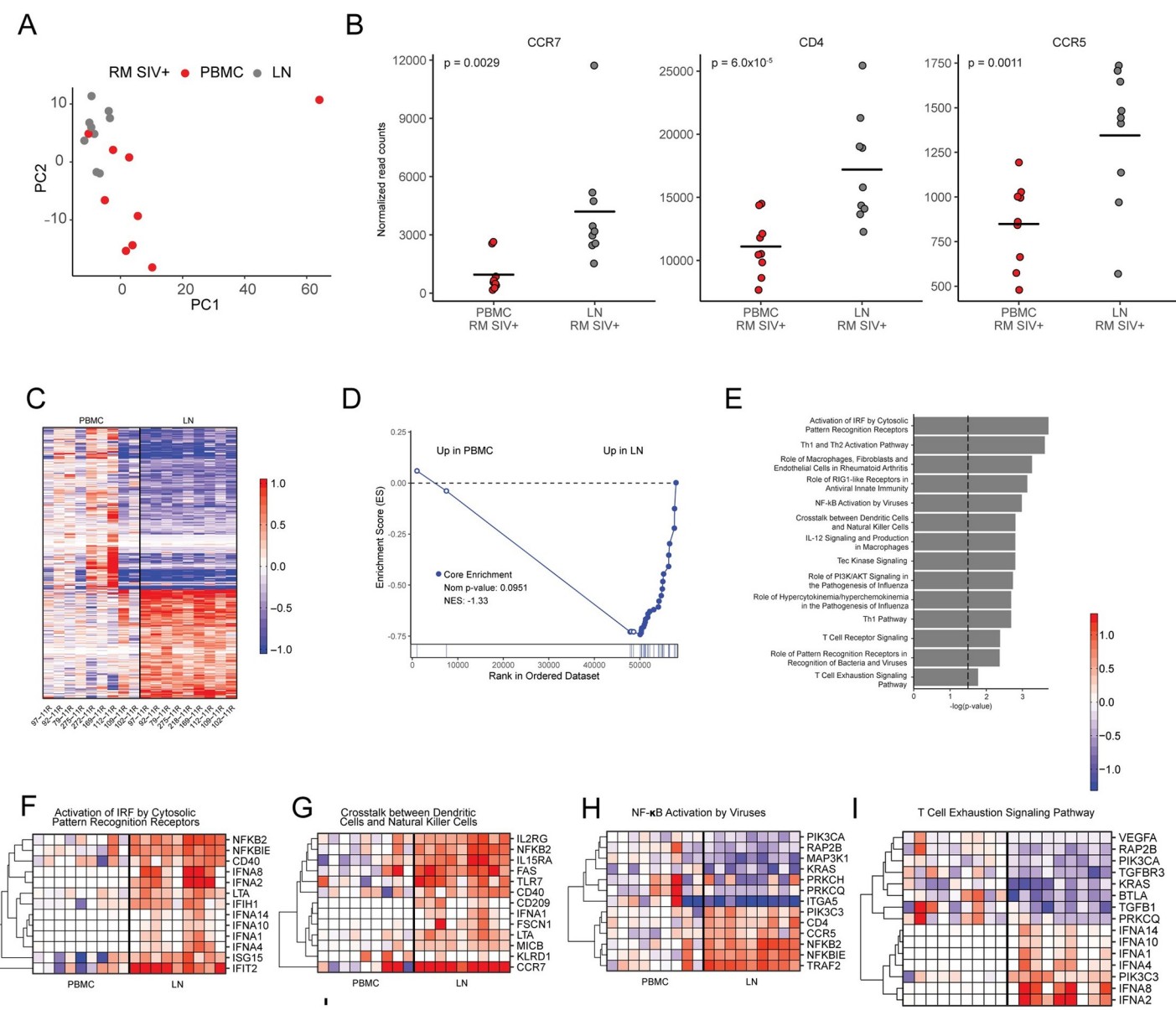

**Fig 4. Pathways in innate and adaptive immunity are upregulated in the LNs but not peripheral blood of SIV-infected RMs. (A)** Principal component analyses (PCA) of the transcriptomes of peripheral blood (red circles) and lymph node (grey circles) pDCs from SIV-infected RMs. (**B**) Dot plots showing normalized read counts of SIV receptor CD4, co-receptor CCR5 and lymph node homing marker CCR7 transcripts in pDCs from the peripheral blood (red circles) or lymph nodes (grey circles) of SIV-infected RMs. Read counts for individual RMs are shown and the mean is indicated by the black bar. P-values from DESeq2 analysis are shown. (**C**) Heat map of the top DEGs (comparing PBMCs and LNs of SIV-infected RMs) from DESeq2, including genes with FDR < 0.05 and FC > 1.5. Data values were transformed using the "rlog" method from DESeq2 and normalization was done by taking the median across PBMC samples and subtracting this value from all observations. Heat map limits were set to -1 to 1. Animal IDs for the individual RMs are included under the heat map. (**D**) GSEA enrichment plot of the Hallmark interferon alpha response (MSIGDB geneset #M5911) for contrasting expression in the peripheral blood vs lymph nodes of SIV-infected RMs. This plot panel also shows NES and nominal p-values. (**E**) Bar chart of -log p-values from IPA. IPA was used with the same genes as the ones included in panel A (PBMC vs LN, FDR < 0.05, FC >1.5) as input. Gene sets from IPA were ordered by -log p-value and the top hits as well as hits of interest are included in the plot. (**F**)—(**J**) Breakout gene sets from the IPA shown in panel **E.** Heat map limits were set at -1.25 to 1.25. Gene set name from the IPA analysis are shown above each plot. In the **F-I** heatmaps, the order of columns matches that of panel **C** and the method of transformation and normalization of the data was identical to that described for panel **C**.

elevated ISGs, the expression was significantly higher in cells derived from LNs. To fully characterize this activation phenotype of LN pDCs, we conducted pathway analysis of genes differentially expressed between pDCs from LN vs blood (**Fig 4D-4J**). Several pathways such as

"Activation of IRF by Cytosolic Pattern Recognition Receptors" and "Role of RIG-I-like Receptors in Antiviral Innate Immunity" were determined to be enriched primarily due to the upregulation of IFNAs and ISGs (**Fig 4E and 4F**). In these pathways, downstream IFN-I signaling molecules were upregulated and not the RIG-I-like/cytosolic pattern recognition receptors themselves. However, pathway analysis demonstrated that the activation of LN derived pDCs was not limited to the IFN/ISG system. Several pathways related to immune activation were also upregulated, notably, "Crosstalk between Dendritic Cells and Natural Killer Cells" and "NFKB activation by viruses", shown in panels of **Fig 4G and 4H**. These are associated with known functions of DCs in linking the innate and adaptive immune responses. To investigate the genes driving enrichment, we visualized gene expression of the top-scoring genes from each pathway. Within the "DC/NK Crosstalk" cluster–we noted consistently elevated expression of CCR7, which directs trafficking to T cell zones in lymph nodes. We also observed elevated expression of CD209/DC-SIGN, a C-type lectin known to trap HIV-1 and several viruses on the surface of DCs. Expression of IL2RG, the common gamma chain, was highly induced in LN pDCs (padj = 0.0017), and the average read counts for this gene were >7000. IL15RA, the high affinity subunit for IL15 binding was also significantly upregulated (padj = $1.7 \times 10^{-4}$) (**Fig 4G** and **S2 Table**). We examined our gene list for other inducible cytokine receptors that utilize IL2RG, and noted that the IL7R transcript levels were elevated in LN pDCs but were just beyond the threshold for significance (padj = 0.07). Expression of the α and β chains of IL2, and the IL9 receptors in pDCs were low, and the IL4R was highly expressed but not significantly different between LN and blood (**S2 Table**). Lastly, we also noted elevated expression of multiple immunostimulatory cell surface molecules: the costimulatory molecule CD40; FAS/CD95 which mediates activation and apoptotic signals and MICB, an MHC class I-like molecule that acts as an activating ligand for NK cells via its interaction with NKG2D. Within the "NF-KB Activation" pathway, we noted elevated expression of the gene encoding the NFKB2 subunit, conversely, we also observed elevated levels of NFKBIE, an inhibitory subunit (**Fig 4H**). This pathway also highlighted the elevated expression of CD4 and CCR5 (**Fig 4H**). Examination of the SIV entry factors showed that there was significantly higher expression of both CD4 and CCR5 in the LN versus PB of SIV-infected RMs, where the fold-change of both surface markers between compartments was comparable (**Fig 4B**). However, like the peripheral blood, CD4 levels are very high (read count 17,208.1) compared to CCR5 (read count 1,344.9). This agrees with flow cytometry data from two uninfected RMs, where MFI of CD4 on LN pDCs is higher than PB pDCs whereas there is no difference in the MFI of the pDC-specific cell-surface marker CLEC4C/BDCA-2 (**S4A Fig**). Interestingly, we noted that a T cell exhaustion signaling pathway was also determined to be significantly enriched (**Fig 4I**), although upregulated genes were mostly represented by IFNAs and other DEGs were downregulated. Collectively, these data (i) demonstrate that LN homed pDCs attain an elevated state of activation that is easily detectable above pDCs in blood; (ii) indicate that pDCs in LN are producing, and responding to, Type I IFN, however they exhibit upregulation of other pathways consistent with activation of NK cells, NFKB activation and T cell exhaustion; (iii) via the elevated expression of the common gamma chain cytokine receptor, IL15RA and IL7RA, indicate that LN pDCs initiate gene expression programs consistent with proliferation and survival. In sum, these data indicate that pDCs in LNs are highly active and have the capacity to be immunostimulatory.

## A compendium of cell surface protein expression for pDCs in SIV infection

One of the advantages of RNA-Seq relative to microarray is that the digital nature of the readout (i.e. read count) correlates with mRNA transcript abundance reasonably well [78,79]. In

this regard, RNA-Seq data can be used to assess the abundance of transcripts, in addition to their ratio of expression relative to another sample. Using a well-defined, purified cellular sub-set, these transcript abundance data can be used to create a "compendium" cataloguing RNA in that subset. Here, we used these data to define a compendium of surface receptors on the pDCs in rhesus macaques. To assay the cell surface genes expressed in RM pDCs, we compiled the read count data from blood-derived pDCs from SIV-infected RMs (n = 9) and ranked them by their median expression and excluded any genes in which the median expression < 100 read counts. We then filtered the list of genes using the "high-confidence" list of putative cell surface proteins in the Cell Surface Protein Atlas/Database found at http://wlab.ethz.ch/cspa/. A plot of the ranked surface markers with key genes of interest indicated and colored according to functional class is shown in **Fig 5A**. The complete set of genes is listed in **S3 Table**. The equivalent plots for ranked surface markers when using the read count data from LN-derived pDCs from SIV-infected RMs (n = 9) is shown in **S5B–S5D Fig**. No significant differences between the compendiums generated from the PB or LN were observed. Cell sur-face proteins associated with antigen presentation (CD74, HLA-DRA, HLA-B) and well estab-lished pDC surface markers such as IL3RA/CD123, LILRA4/ILT7, FLT3/CD135, CLEC4C/CD303/BDCA-2 were highly expressed (**Fig 5B**). Interestingly, the levels of CD4, the primary entry receptor for HIV/SIV was one of the most highly expressed genes detected, while the HIV/SIV co-receptor CCR5 had modest expression. Within the chemokine receptor family, we detected CCR2, CCR5, CCR7, CXCR3 and CXCR4, all others were below our defined threshold of detection (median > 100 reads). In SIV-infected RMs and using flow cytometry, pDCs have been reported to express CD40 but lack expression of the other co-stimulatory markers CD86 and CD83 [80]. We observed that expression of all three markers by mRNA was only marginally above our defined limit of detection. The mRNA levels for both IFNAR1 and IFNAR2 subunits was detected at high levels.

We also investigated the expression of several immunoregulatory and checkpoint inhibitor molecules known to influence T cell activation during HIV infection. Consistent with prior lit-erature reports, we noted that the absolute expression of both the IL10RA and IL10RB subunits was exceptionally high, with both molecules exceeding 1000 read counts. Expression of BTLA1 was barely above our 100 read count cut-off (read count 116). Expression of the checkpoint blockade molecule PD1 (gene symbol PDCD1) was similarly barely above our defined thresh-old (read count 130); and PDL1/CD274 was only slightly higher at median read count of 402. In contrast, we observed remarkably high expression of the inhibitory receptor LAIR1 (read count 16,818), at levels close to those observed for the MHC molecules. Cross-linking of LAIR1 on healthy human PB pDCs has been shown to inhibit IFN-α production in response to engagement of TLR9 [81]. We also observed significantly high levels of mRNA for the exhaustion marker TIGIT (read count 4,547). TIGIT is highly expressed on resting and acti-vated NK cells[82] and has been shown to mark exhausted CD8+ T cells in HIV and SIV infec-tion [83], and is enriched on CD4+ T cells harboring latent virus in ART treated patients [84]. In contrast, the expression of TIM3, an exhaustion marker that has been shown to be highly expressed on HIV-1 infected human pDCs [85,86], was negligible in PB and LN pDCs from SIV-infected RMs.

To validate the predictive accuracy of our summed RNA-Seq dataset, we performed flow cytometry on an additional set of SIV-infected RMs for TIGIT and PD1 at day 35 post-infec-tion on pDCs and CD8+ T cells from the PB. Representative stains of three SIV+ RMs is shown in **Fig 5C** (staining of n = 6 RMs is shown in **S6 Fig**). As previously demonstrated, both TIGIT and PD1 were expressed at high levels on CD8+ T cells. In concordance with the mRNA ranked expression data, the level of TIGIT protein expressed on pDCs at MFI's similar to that of CD8+ T cells and CD4+ T cells (**Figs 5C and S6**). In contrast, in agreement with the

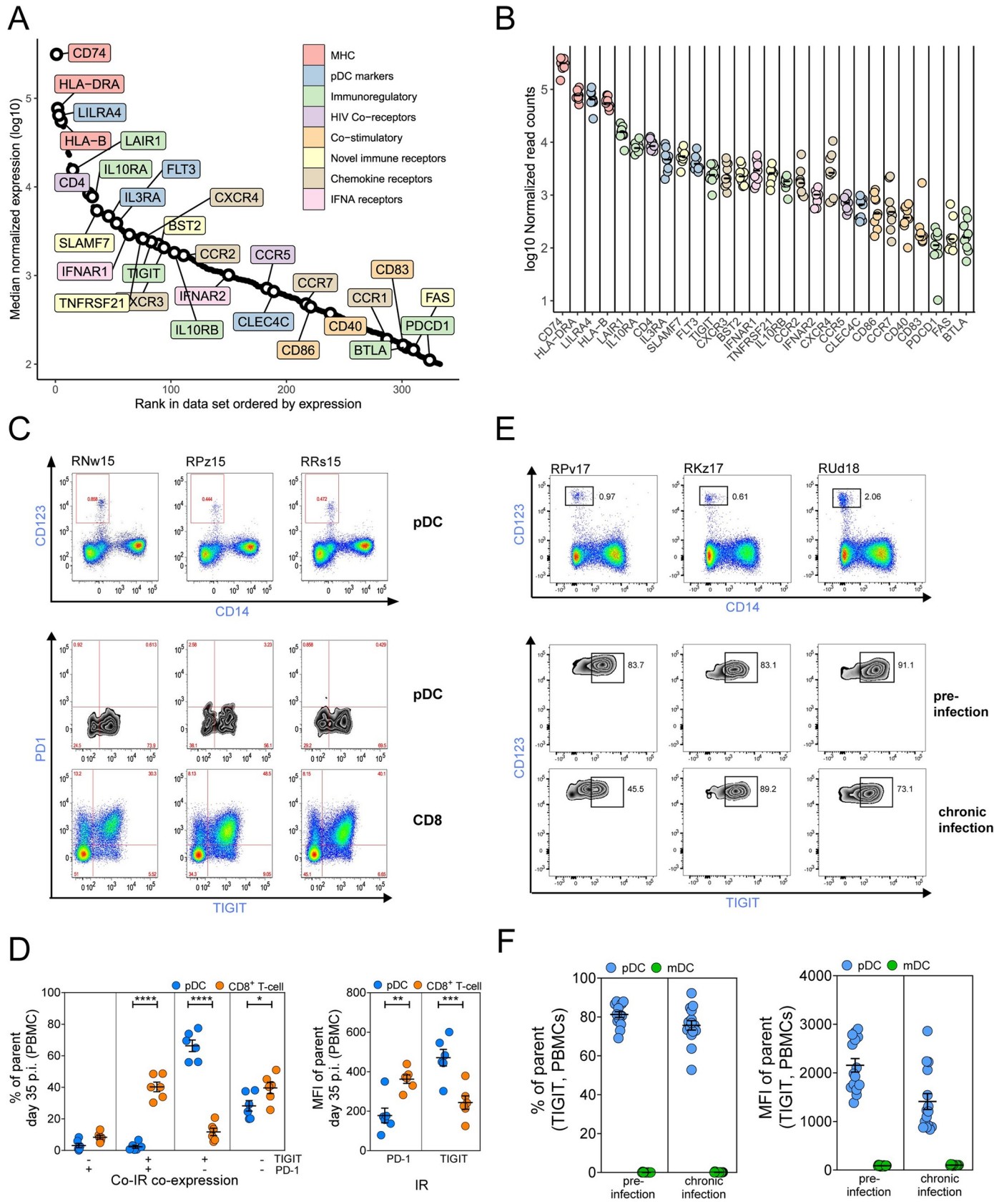

**Fig 5. Cell surface protein expression compendium for pDCs in SIV-infected RMs. (A)** Expression (log10) of surface protein genes with the highest expression (median expression across all samples >100) in SIV+ RM PBMC samples (n = 9). Preliminary list of surface proteins obtained from http://wlab.ethz.ch/cspa/ with five additional genes of interest included (CLEC4C, LILRA4, CCR5, CCR2, CXCR4). Labelled genes of interest are color-coded according to their annotated immune function. Note that this plot contains data aligned to human reference (build version GRCh38) because of missing annotations in MacaM, hence the appearance of non-RM MHC class I genes. **S4A Fig** shows the same data aligned to MacaM. **(B)** Log10 normalized read counts for each cell surface protein gene, where individual RMs are shown with circles and the mean indicated by the black bars. **(C)** Expression of PD-1 and TIGIT on the surface of PB pDCs (CD123 + CD14-) or CD8+ T cells in three SIV-infected RMs (not sampled for RNA-Seq) as determined by flow cytometry. Representative gating of three animals for the pDC population and PD-1, TIGIT expression on pDCs and CD8+ T cells are shown. **(D)** Proportions of pDCs (blue circles) or CD8+ T cells (orange circles) individually expressing or co-expressing TIGIT or PD-1 at 35 days post-infection with SIV (left panel). Mean fluorescence intensity (MFI) of TIGIT or PD-1 detected on pDCs (blue circles) or CD8+ T cells (orange circles) at 35 days post-infection with SIV (right panel). IR = inhibitory receptor. The values for six RMs are shown and mean +/- SEM is indicated by the black line and error bars. **(E)** Representative flow cytometry plots showing gating of the PB pDC population (CD14- CD123+) and expression of TIGIT on this population in three RMs before and after infection with SIV. **(F)** Proportions of pDCs (blue circles) and mDCs cells (green circles) expressing TIGIT pre-infection and at 42 days post-infection (chronic infection) (left panel). MFI of TIGIT detected on pDCs (blue circles) in comparison to mDCs (green circles) pre-infection and at 42 days post-infection (right panel). Statistical significance in **(D)** was determined by Two-way repeated measures ANOVA followed by Bonferroni's multiple comparisons test. ns = not significant p > 0.05, $^*$ p ≤ 0.05, $^{**}$ p ≤ 0.01, $^{***}$ p ≤ 0.001, $^{****}$ p ≤ 0.0001.

RNA-Seq data, PD1 protein was not found at detectable levels on pDCs by flow cytometry, but observed at high levels on both CD4+ and CD8+ T cells (**Figs 5C and S6**). Co-inhibitory receptor expression analysis showed that pDCs were predominantly single positive for TIGIT (~65%) or double negative for TIGIT and PD-1 (~30%) whereas CD8+ T cells had similar proportions of double negative and double positive populations (~40%) and single positive populations (~10%) (**Fig 5D**). The level of TIGIT protein on pDCs from PB as shown by MFI was significantly higher than that of CD8+ T cells, whereas PD-1 was undetectable on pDCs in PB (**Fig 5D**). To investigate the kinetics of surface expression of TIGIT on PB pDCs before and after SIV infection, we performed flow cytometry on an independent set of 15 RMs. We observed a high frequency of TIGIT+ pDCs prior to infection that was maintained in early chronic infection (Day 42) (**Fig 5E**, representative staining of 3 RMs and **Fig 5F**).

Lastly, we also assessed expression levels of LAIR1 on pDCs by flow cytometry of whole blood from eight SHIV-infected RMs at 16 weeks post-infection. The MFI of LAIR1 on PB pDCs was the highest out of all immune cell types detected which included CD4+ T cells, CD8 + T cells, CD14+ monocytes, and conventional CD11c+ dendritic cells (**S7 Fig**).

Overall, these data demonstrate that our ranking approach to identify molecules on the cell-surface of pDCs was well supported by flow cytometry data, the somewhat arbitrary threshold of 100 read count as a limit of detection correlated well with flow cytometry detection. Additionally, these data identified several novel surface molecules previously not demonstrated on pDCs.

A compendium was also generated for transcription factors expressed by PB pDCs (**S8A and S8B Fig**, **GRCh38 and MacaM annotations**) and LN pDCs (**S8C and S8D Fig**, **GRCh38 and MacaM annotations**) in SIV-infected RMs, again using the median read count of 100 threshold. The full list of genes is listed in **S4 Table**. The Interferon regulatory factors (IRF) and signal transducer and activator of transcription (STAT) protein family dominate this compendium. As expected, two very highly expressed TFs are those that establish pDC fate, IRF8 and TCF4 (E2-2) with an interactor of the latter, BCL11A, which maintains pDC fate, highly expressed as well [87–91].

## Discussion

In this study we have used transcriptional profiling to examine the function of plasmacytoid dendritic cells in chronic SIV infection in two species of NHPs. Our main observations were that (i) pDCs in non-natural hosts maintain high levels of ISGs in chronic infection; in contrast, ISG signatures are lacking in pDCs from SIV-infected natural host species; (ii) pDCs in lymphoid tissues maintain a highly immunostimulatory transcriptional profile relative to

pDCs in blood; (iii) RNA-Seq can be a reliable tool to survey the composition of surface proteins in a purified immune cellular subset and (iv) the exhaustion molecules LAIR1 and TIGIT were found to be highly expressed on pDCs in SIV infection. Lastly, one of the most important findings in this study was that during chronic SIV infection, peripheral lymph node homed pDCs produce Type I IFN transcripts and protein, in particular various subtypes of IFNA. Collectively, these data indicate that pDCs have high potential for driving immune activation in chronic SIV and HIV infection.

One of primary questions we sought to answer in this study was to test if pDCs contributed to the production of the Type I IFN response in chronic SIV infection. Persistently elevated levels of ISG expression are a consistently observed feature of chronic HIV and SIV infection and has been strongly linked to the detrimental hyperactivation of the immune system [92]. pDCs are the predominant producer of Type I IFN and have been estimated to be responsible for approximately 99% of IFNA production [93]. The ability of pDCs to produce Type I IFNs during acute SIV infection is broadly accepted, as IFNA and IFNB protein has been detected in LN pDCs by immunohistochemistry [63] and IFNA protein by direct ex vivo flow cytometry [64]. However, prior NHP studies have suggested that pDCs may not contribute to IFNA production once SIV infection shifts to the chronic phase: work by Brown and colleagues [94] showed that pDCs in LN were largely apoptotic at 14 days post-infection. Other work in SIV-infected cynomolgus macaques was able to detect intracellular IFNA protein levels in pDCs from blood and LN using direct ex vivo flow cytometry at early time points after SIV infection (day 10), but did not detect IFNA at 30 d.p.i. or later time points [64]. In the mouse model, Colonna and colleagues depleted pDCs using CLEC4C-DTR transgenic mice where pDCs are eliminated after injection of diphtheria toxin and demonstrated pDCs were dispensable for control of chronic herpesvirus infections [95].

Here, using RNA-Seq, we have detected robust levels of IFNA transcripts in seven out of nine monkeys in pDCs isolated from LNs at the chronic stage of infection. Our supporting data demonstrate that (i) detection of IFN-α protein is exclusively co-localized to CD303 + pDCs in LNs, providing protein validation of the RNA-Seq observation; (ii) pDCs are the primary producers of IFNA in response to TLR7 ligands and SIV, and (iii) by enriching pDCs in PBMCs cultures, we were able to significantly increase IFNA production in vitro; (iv) isolated splenic pDCs from uninfected RMs did not have detectable IFNA RNA–collectively these suggest that the IFNA transcripts observed in chronically infected RMs were not from a contaminating cell source, and were also detected at the protein level. The discrepancy between our data, in which we clearly observe IFNA production in chronic SIV infection, and those of Bruel et al. [64], which failed to detect IFNA protein, is not entirely clear. Our study used Indian origin RMs, whereas theirs utilized cynomolgus macaques. A more likely explanation may be differences in the number of cells analyzed–we were able to obtain thousands of pDCs due to sampling at necropsy, compared to hundreds of pDCs available during longitudinal sampling. An additional difference may also be the relative sensitivity of the assays used to detect IFNA: (i) in RNA-Seq, the signal (read count) is summed from all cells, whereas by intracellular flow cytometry, the signal is averaged across all cells; (ii) the library preparation methodology we employed uses an initial PCR amplification step that helps to enrich low abundance transcripts, and (iii) RNA-Seq quantitates reads from all IFNA species, across the entire length of the transcript, whereas intracellular flow cytometry is limited to a single epitope. We did not observe IFNA in our negative control splenic pDC samples from uninfected animals, which indicates that the production of IFNA was specific to SIV infection, and not simply a feature of tissue derived pDCs. It is important to note that our findings in the NHP model are consistent with observations in ART-naïve, HIV-infected individuals, in which pDCs homing to LNs were found to produce high levels of IFNA [96].

Our observation that IFNA could be found in LN derived pDCs, but not in blood, suggests that activated pDCs are enriched in lymphoid tissues. This is consistent with trafficking studies, in which pDCs have been shown to be more frequent in the LN, vaginal and rectal mucosa after SIV and HIV infection where they express elevated levels of co-stimulatory markers, apoptotic markers or beta chemokines [59,60,94]. This observation is also consistent with the findings of Sabado et al [57], who studied human pDCs during HIV infection and showed only subtle differences in microarray comparisons of blood pDCs from uninfected and primary HIV infected donors [57]. When we contrasted the transcriptome of pDCs from LNs with those from blood, we observed modulation of several immune-related pathways in LNs beyond Type I IFNs and ISGs, notably upregulation of NFKB signaling, DC mediated activation of NK cells and T cell exhaustion. It should be noted that the transcriptional profile of blood pDCs from infected animals had several immune-related pathways elevated relative to cells from uninfected animals–thus the observation that, during chronic SIV infection, the activation state of LN pDCs was easily distinguishable from blood indicates that these cells were hyperactivated. Interestingly, we also observed that pDCs from LNs had significantly elevated levels of cell-associated SIV RNA transcripts detected by RNA-Seq. Work by Koup and colleagues demonstrated that pDCs could support a moderate level of HIV replication when infected in vitro [97], however, in vivo, others have failed to detect HIV provirus in pDCs from peripheral blood of ART-naïve PLWH [98]. Our observations of cell-associated SIV transcripts, cannot address if pDCs harbour SIV replication in tissues. However, the increased activation profiles of the LN pDCs may be partially due to being exposed to higher levels of SIV in the LN, but additional work will be required to test this formally.

We also observed significant upregulation of CCR7 and CCR5 in LN pDCs, suggesting that LN pDC migrate to T cell zones. These data suggest that in SIV-infection, hyperactivated pDCs capable of IFNA production may traffic to T cell zones and provide immunostimulatory signals to T cells. Indeed, recent work in mice has shown that pDCs migrate to LN and interact with XCR+ DCs to prime CD8+ T cells during a primary immune response, and that this activity was orchestrated via CCR5 [99]. Our observation of T cell exhaustion pathways enriched in pDCs are consistent with a model in which hyperactive pDCs localize in T cell areas of LNs and contribute to T cell dysfunction and exhaustion, although this needs to be tested by functional studies in vivo. Therapeutic targeting of IFN has received significant recent interest to ameliorate HIV-related immune activation and to enhance strategies to reduce the latent HIV reservoir [29]. IFN-blockade in HIV-infected humanized mice was able to reduce levels of latently infected CD4+ T cells; this activity was attributable to reducing the impact of checkpoint blockade inhibitors on CD8+ T cells and enhancing CD8+ T cell responses [8,9]. Efficient inhibition of the IFN system in vivo is technically challenging as there are 13 subtypes of IFNA in humans. Further, direct blockade of the IFN-I system has the disadvantage of being pleiotropic, and may confer undesired side effects, such as immunodeficiency. In this respect, pDCs may offer a more tractable target to reduce IFN-based inflammation: they are a relatively infrequent population, that, based on the data presented here, may exert substantial pro-inflammatory stimuli on T cells by virtue of their localization and high capacity for immunostimulatory signaling. Novel reagents inhibiting pDC activity have shown promise in clinical trials in reducing the severity of symptoms in lupus [100]. Our data suggest that pDCs may be a viable drug target to reduce HIV-associated immune activation and related sequelae, such as T cell exhaustion.

Our study contains some drawbacks. The median age of the SIV-infected SMs was 17 years compared to 4 years for the RMs and the duration of infection of SMs was 11 years compared to 7.5 months for the SMs. In our prior work, where younger SMs, aged 5–11, were infected for a shorter amount of time, we observed a similar lack of an ISG response in the periphery at

180 days post-infection [19], which matches the results from the pDCs and mDCs in the current study. Age-dependent changes in innate immunity have been described, however the lack of an ISG response in the chronically infected SMs in the current study agrees with the observed rapid shutdown of IFN post-infection in nonpathogenic NHP hosts of SIV as reported by us and others [19–23]. It is not possible to repeat this study to control for age and duration of infection due to the moratorium on de novo SIV infections of sooty mangabeys. The comparison of pDCs between RMs and SMs was relatively underpowered at n = 4, thus we relied on pathway-based analysis and not classical gene-centric inferential statistics. Nevertheless, our data recapitulated several findings shown by orthogonal methods (e.g. CCR5 expression) suggesting that our findings were accurate. Our observation of IFNA transcripts in LN pDCs could potentially be due to contaminating cells within the pDC preparations; however, our supporting data demonstrate that pDCs are virtually the exclusive producers of IFNA. Another weakness in our study was that we observed IFNA in the LNs of seven of nine animals. The animals in which we failed to detect IFNA had substantially lower pDC yields compared to the others, and it is possible that this low cell number hampered detection of IFNA transcripts–indeed, detection of IFNA, even at peak SIV replication is remarkably difficult. However, it is important to note that even in animals in which IFNA was not detected, a strong upregulation of the ISG system was observed. However, we observed that the RNA of ISGs downstream of Type I IFN were hyperelevated in LN pDCs, which suggests that bioactive IFNA was present.

One of the surprising findings in this study was the high, constitutive expression of the inhibitory immune checkpoint molecule TIGIT on pDCs. TIGIT has been reported to be highly expressed on NK cells and CD8+ T cells and to play a role in the regulation of T cell activation and exhaustion and NK cell dysfunction [101]. We observed no significant differences in the levels of TIGIT mRNA in pDCs between SIV- and SIV+ samples, nor between blood and LNs derived cells. While our data did not address the function of TIGIT on pDCs, the significant role that pDCs play in CD8+ T cell and NK activation suggests that the immunoregulatory mechanisms of TIGIT signaling would allow for a downmodulation of all three subsets in a concerted manner. Pharmacological blockade of TIGIT is currently under intense investigation as a therapy to reverse HIV latency and enhance anti-HIV activity by CD8+ T cells and NK cells; our data suggest that an added mechanism of action of this drug may be to interfere with pDC activity.

Immune activation remains an important barrier to health to PLWH, even in the setting of highly effective antiretroviral therapy. Moreover, immune activation may promote immune dysfunction such as T cell exhaustion that further complicate therapies aimed at reducing the latent reservoir. In this study, we investigated the potential of pDCs to drive HIV driven immune activation using comparative studies of natural and non-natural NHP host species of SIV infection and ex vivo analysis of immune tissues during chronic SIV infection. Our data shows that LN-derived pDCs produce IFNA at late stages of chronic SIV infection, and are hyperactivated, in contrast to prior reports. These data help to elucidate the role of pDCs in chronic lentiviral infection. They also identify pDCs as a rational therapeutic target to ameliorate HIV-driven immune activation and restore effective T cell-based immunity.

## Supporting information

**S1 Fig. Flow Cytometry gating strategy for pDCs.** pDCs are defined as live CD3- CD20- HLA-DR+ CD14- CD11c- CD123+ leukocytes and shown with the blue circle in the final scatter plot. mDCs are defined as live CD3- CD20- HLA-DR+ CD14- CD11c+ CD123- leukocytes and shown with the black circle in the final scatter plot. Non-pDCs used in the ELISpot assay

were sorted using the same strategy and are indicated by the purple gate (CD3- CD20-
HLA-DR+ CD14- CD123-).
(TIF)

**S2 Fig. Expression of ISGs in mDCs. (A)** Principal component analyses (PCA) of the tran-
scriptomes of uninfected (circles) and infected (squares) pDCs and mDCs from RMs and SMs.
RM pDCs are in red, RM mDCs are in green, SM pDCs are in blue and SM mDCs are in pur-
ple. Heat maps of 17 genes from an ISG signature panel representative of Type I IFN signaling
for uninfected and SIV-infected RMs (**B**) and uninfected and SIV-infected SMs (**C**). The val-
ues shown were first transformed using the "rlog" method from DESeq2, data were normalized
by subtraction of the median (on a per-gene basis) of the baseline samples (i.e. SIV- RM and
SIV-SM) from each datapoint; RMs data was normalized by the SIV-RM median and SM data
by the SIV-SM median. The color scale of the heatmap is set to maximal at fold-changes of -1.5
and 1.5. Normalized read counts (log10) for the 17 ISGs is shown in the bottom graphs. Dots
show median normalized read counts (log10) of n = 4 for each experimental group.
(TIF)

**S3 Fig. Minimal expression of TLRs other than TLR7 and TLR9 on pDCs.** Normalised read
counts for the cell surface (TLR1, 2, 4, 5, 6, 10) and intracellular (TLR3, 8) pattern recognition
receptors in pDCs. All plots show values for individual RMs and SMs and black line indicates
mean, n = 4 for each condition.
(TIF)

**S4 Fig. High levels of CD4 expression on pDCs and detection of SIV reads in LN pDCs.** (**A**)
Median fluorescence intensity (MFI) of CLEC4C/BDCA-2 or CD4 expressed on the surface of
PB pDCs (blue circles) and mDCs (green circles) from 15 uninfected RMs (left) detected via
flow cytometry. Median fluorescence intensity (MFI) of CLEC4C/BDCA-2 or CD4 expressed
on the surface of PB (red) or LN (gray) pDCs from two uninfected RMs (right) detected via
flow cytometry. Levels of CLEC4C/BDCA-2 or CD4 were measured on the CD11c- CD123
+ or CD11c+ CD123- populations gated as shown in **S1 Fig**. (**B**) Raw read counts of SIV tran-
scripts mapped to SIVmac239 in pDCs isolated from the blood (red) or lymph node (grey) of
matched and unmatched SIV+ RM (left panel). Read counts are shown for individual RMs
and the black bar indicates mean. Relationship between input pDC cell number and normal-
ized SIV read count (right panel).
(TIF)

**S5 Fig. Cell surface protein compendium for pDCs in SIV-infected RMs aligned to MacaM
and from LN-derived pDCs. A**. Expression (log10) of surface protein genes with the highest
expression (median expression across all samples >100) in SIV+ RM PBMC samples (n = 9). Pre-
liminary list of surface proteins obtained from http://wlab.ethz.ch/cspa/ with five additional genes
of interest included (CLEC4C, CCR5, CCR2, CXCR4). Labelled genes of interest are color-coded
according to their annotated immune function. **B-D**. The equivalent surface protein gene expres-
sion plots for LN-derived pDCs, **B** and **C** were aligned to GRCh38, **D** was aligned to MacaM.
(TIF)

**S6 Fig. Expression of TIGIT and PD-1 on pDCs, CD4+ and CD8+ T cells determined by
flow cytometry.** pDCs gated as CD14- CD123+ cells from live CD3- CD20- HLA-DR+ lym-
phocytes (top panel). Flow cytometry plots showing expression of PD-1 and TIGIT on periph-
eral blood pDCs, CD4+ or CD8+ T cells of six SIV-infected RMs at 35 days post-infection
(bottom 3 panels).
(TIF)

**S7 Fig. Expression of LAIR1 on surface of immune cell subsets determined by flow cytometry.** Median Fluorescence Intensity (MFI) of LAIR1 on total T cells (CD3+), CD4+ T cells, CD8+ T cells, B cells (CD20+), monocytes (CD14+), conventional dendritic cells (CD11c+) and plasmacytoid dendritic cells (CD123+) from eight SHIV-infected RMs at 16 weeks post-infection. Statistical comparisons were made using a paired T test.
(TIF)

**S8 Fig. Transcription factor protein compendium for pDCs in SIV-infected RMs.** Expression (log10) of transcription factor genes with the highest expression (median expression across all samples >100) in SIV+ RM PB pDC samples (n = 9). Labelled genes of interest are color-coded according to their annotated functional families. Alignments to both GRCh38 (**A**) and MacaM (**B**) are shown. The equivalent plots for SIV+ RM LN-derived pDC samples (n = 9), with alignment to GRCh38 (**C**) and MacaM (**B**).
(TIF)

**S1 Table. Features of the rhesus macaques and sooty mangabeys used in the transcriptomics and immunofluorescence microscopy components of the study.**
(XLSX)

**S2 Table. List of differentially expressed genes between peripheral blood and lymph node pDCs of SIV infected rhesus macaques.**
(XLSX)

**S3 Table. List of genes above the threshold of 100 read counts included in the compendium of cell surface protein expression for pDCs in SIV infection of rhesus macaques.**
(XLSX)

**S4 Table. List of genes above the threshold of 100 read counts included in the compendium of cell transcription factor expression for pDCs in SIV infection of rhesus macaques.**
(XLSX)

## Acknowledgments

Veterinary support was provided by Joyce Cohen and the staff of the Yerkes Animal Resources Division, and sampling of NHP blood and tissues was organized by Stephanie Ehnert and performed by the staff of the Yerkes Research Services Division, and Tracey Meeker in the Yerkes Colony Management unit. Illumina sequencing was conducted by Nirav Patel in the Yerkes NHP Genomics Core. Plasma SIV viral loads were conducted in the Translational Virology Core of the Center for AIDS Research at Emory University supervised by Dr. Thomas Vanderford.

## Author Contributions

**Conceptualization:** Michelle Y.-H. Lee, Amit A. Upadhyay, Hasse Walum, Steven E. Bosinger.

**Data curation:** Michelle Y.-H. Lee, Amit A. Upadhyay, Gregory K. Tharp.

**Formal analysis:** Michelle Y.-H. Lee, Amit A. Upadhyay, Hasse Walum, Chi N. Chan, Kyndal L. Goss, Gregory K. Tharp, Vijayakumar Velu, Jacob D. Estes, Steven E. Bosinger.

**Funding acquisition:** Steven E. Bosinger.

**Investigation:** Michelle Y.-H. Lee, Chi N. Chan, Sydney A. Nelson, Ernestine A. Mahar, Diane G. Carnathan, Barbara Cervasi, Kiran Gill, Elizabeth R. Wonderlich, Vijayakumar Velu.

**Methodology:** Michelle Y.-H. Lee, Chi N. Chan, Reem A. Dawoud, Christine Grech, Justin L. Harper, Kirti A. Karunakaran, Barbara Cervasi, Kiran Gill, Elizabeth R. Wonderlich.

**Project administration:** Reem A. Dawoud, Ernestine A. Mahar, Steven E. Bosinger.

**Resources:** Simon M. Barratt-Boyes, Mirko Paiardini, Guido Silvestri, Jacob D. Estes, Steven E. Bosinger.

**Supervision:** Steven E. Bosinger.

**Validation:** Michelle Y.-H. Lee, Chi N. Chan.

**Visualization:** Michelle Y.-H. Lee.

**Writing – original draft:** Michelle Y.-H. Lee, Amit A. Upadhyay, Hasse Walum, Steven E. Bosinger.

**Writing – review & editing:** Steven E. Bosinger.

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
