## [Decision Letter · Decision Letter 0]

30 Nov 2020

Dear Dr. Bosinger,

Thank you very much for submitting your manuscript "Tissue-specific transcriptional profiling of plasmacytoid dendritic cells reveals a hyperactivated state in chronic SIV infection" for consideration at PLOS Pathogens. As with all papers reviewed by the journal, your manuscript was reviewed by members of the editorial board and by several independent reviewers. In light of the reviews (below this email), we would like to invite the resubmission of a significantly-revised version that takes into account the reviewers' comments.

We cannot make any decision about publication until we have seen the revised manuscript and your response to the reviewers' comments. Your revised manuscript is also likely to be sent to reviewers for further evaluation.

Sincerely,

Daniel C. Douek

Associate Editor

PLOS Pathogens

Richard Koup

Section Editor

PLOS Pathogens

Kasturi Haldar

Editor-in-Chief

PLOS Pathogens

orcid.org/0000-0001-5065-158X

Michael Malim

Editor-in-Chief

PLOS Pathogens

orcid.org/0000-0002-7699-2064

Reviewer's Responses to Questions

**Part I - Summary**

Reviewer #1: In their manuscript, Lee and colleagues perform a cross-NHP species transcriptomic analysis of pDCs to evaluate their role(s) in ongoing HIV-induced immune activation. The investigator’s conclude in the pathogenic rhesus species, pDCs develop a hyperactivated state in lymph nodes during SIV infection. The study is mostly well done and very well-written, but there are some experimental and interpretive concerns that should be addressed.

Reviewer #2: Lee et al studied pDC in SIV rhesus macaque infection to further understand their role. They find that pDC appear to be recruited to lymph nodes during infection and have an enhanced activation state as demonstrated by RNA-seq. The authors claim that the pDC are driving T cell activation and exhaustion, although this is not formally demonstrated in functional assays. Data are shown that confirm that pDC are major IFNa producers. The authors also suggest that pDCs themselves are exhausted. It is unclear to this reviewer whether they are indicating that pDC in blood are exhausted or those both in LN and blood. If the pDC in LN in blood are exhausted, then why is IFNa being produced. This needs to be clarified. The manuscript has used RNA-seq technology to further delineate the role of pDC in SIV (and thus test in HIV) infection. This methodology may allow easier investigation of such a difficult to study cell population ex vivo.

**Part II – Major Issues: Key Experiments Required for Acceptance**

Reviewer #1: The SMs were clearly infected much longer than rhesus. How was this evaluated as a potential confounding factor? It may be difficult to make a direct comparison at all here.

A similar question applies for the fact that the SMs were likely much much older than the rhesus used. This could have a major impact on ISG and other gene expression and should be clarified for the infected and uninfected animals in both species. This may also require some additional analyses.

It is unclear to this reviewer, how it was determined these are tissue-resident pDCs in LN, rather than pDC that may have migrated there or are blood contaminating. This should be rephrased unless specific confirmation of tissue-residency markers (CD69, CD49s, etc) were used. This does not undermine the disparate profiles of LN vs. circulating pDC, but given recent changes in how tissue-resident populations are identified, wording should be careful.

It would also be useful to confirm IFN production in the peripheral and LN pDCs by ICS or ELISPOT or ELISA to verify the conclusions by transcripts.

Were ISGs in non-pDCs upregulated correlating within pDCs? Since the pDC may be a primary source this would seem a natural analysis rather than necessarily looking at ISGs in the pDCs themselves.

The authors make the argument that hyperactivated pDCs may have a role in lentivirus disease. Were no virologic measurements performed? Particularly in LN.

Reviewer #2: No functional data is shown that pDC are indeed exhausted or whether they drive exhaustion of other cell types.

**Part III – Minor Issues: Editorial and Data Presentation Modifications**

Reviewer #1: (No Response)

Reviewer #2: Clarify if the macaques in figure one that are SIV infected are on cART or not.

Please add the p value for the uninfected counterparts on line 347, and in line 359

Fig 3d, please show error bars.

Also In fig 3 Lee et al shows shows constitutive expression in infected animals of RNA, is this true for protein expression?, this condition could be added in fig 3d, ie add pdc from uninfected animals at the same concentration without stimulation.

Do we know whether the pDC in LN versus PB or both contain SIV virus as others have shown? If so, can this explain some of the findings?

Is it confirmed that pDC in one compartment are CCR7 surface positive in one compartment over another?

TIM-3 has been associated with pdC exhaustion in mice and humans, what was TIM-3 levels in these pDCs.

Exhaustion markers TIGIT is elevated in pdC from SIV infected animals but this reviewer does not see expression of pDC from uninfected animals. In fig 5c, it is unclear whether the pDC are from periphery or LN.

PLOS authors have the option to publish the peer review history of their article (what does this mean?). If published, this will include your full peer review and any attached files.

Reviewer #1: No

Reviewer #2: No
---

## [Decision Letter · Decision Letter 1]

28 May 2021

Dear Dr. Bosinger,

We are pleased to inform you that your manuscript 'Tissue-specific transcriptional profiling of plasmacytoid dendritic cells reveals a hyperactivated state in chronic SIV infection' has been provisionally accepted for publication in PLOS Pathogens.

Best regards,

Daniel C. Douek

Associate Editor

PLOS Pathogens

Richard Koup

Section Editor

PLOS Pathogens

Kasturi Haldar

Editor-in-Chief

PLOS Pathogens

orcid.org/0000-0001-5065-158X

Michael Malim

Editor-in-Chief

PLOS Pathogens

orcid.org/0000-0002-7699-2064

Reviewer Comments (if any, and for reference):

Reviewer's Responses to Questions

**Part I - Summary**

Reviewer #1: (No Response)

Reviewer #2: confirms pDCs are inducing activation in this model

**Part II – Major Issues: Key Experiments Required for Acceptance**

Reviewer #1: (No Response)

Reviewer #2: (No Response)

**Part III – Minor Issues: Editorial and Data Presentation Modifications**

Reviewer #1: (No Response)

Reviewer #2: (No Response)

PLOS authors have the option to publish the peer review history of their article (what does this mean?). If published, this will include your full peer review and any attached files.

Reviewer #1: No

Reviewer #2: No

---

## [Editor Report · Acceptance letter]

22 Jun 2021

Dear Dr. Bosinger,

We are delighted to inform you that your manuscript, "Tissue-specific transcriptional profiling of plasmacytoid dendritic cells reveals a hyperactivated state in chronic SIV infection," has been formally accepted for publication in PLOS Pathogens.

Best regards,

Kasturi Haldar

Editor-in-Chief

PLOS Pathogens

orcid.org/0000-0001-5065-158X

Michael Malim

Editor-in-Chief

PLOS Pathogens

orcid.org/0000-0002-7699-2064